# Structural basis of SIRT7 nucleosome engagement and substrate specificity

Carlos Moreno-Yruela [1,4] ✉, Babatunde E. Ekundayo [2,3,4], Polina N. Foteva [1], Dongchun Ni [2,3], Esther Calvino-Sanles [1], Henning Stahlberg [2,3] & Beat Fierz [1] ✉

Chromatin-modifying enzymes target distinct residues within histones to finetune gene expression profiles. SIRT7 is an NAD⁺-dependent deacylase often deregulated in cancer, which deacetylates either H3 lysine 36 (H3K36) or H3K18 with high specificity within nucleosomes. Here, we report structures of nucleosome-bound SIRT7, and uncover the structural basis of its specificity towards H3K36 and K18 deacylation, combining a mechanism-based cross-linking strategy, cryo-EM, and enzymatic and cellular assays. We show that the SIRT7 N-terminus represents a unique, extended nucleosome-binding domain, reaching across the nucleosomal surface to the acidic patch. The catalytic domain binds at the H3-tail exit site, engaging both DNA gyres of the nucleosome. Contacting H3K36 versus H3K18 requires a change in binding pose, and results in structural changes in both SIRT7 and the nucleosome. These structures reveal the basis of lysine specificity, allowing us to engineer SIRT7 towards enhanced H3K18ac selectivity, and provides a basis for small molecule modulator development.

Sirtuins are ubiquitous regulators of protein function through NAD⁺-dependent cleavage of $N^\epsilon$-acyl-lysine post-translational modifications (PTMs)[1,2], and are deeply involved in chromatin regulation via histone deacetylation[3]. The sirtuin family in mammals consists of seven members corresponding to class I (SIRT1–3), class II (SIRT4), class III (SIRT5), and class IV (SIRT6, SIRT7)[4]. Four of these enzymes—SIRT1, SIRT2, SIRT6, and SIRT7—erase histone lysine acetylation (Kac) to regulate chromatin structure and gene transcription, and they have isozyme-specific roles as epigenetic regulators of disease pathways in cancer, neurodegeneration, and cardiometabolic disorders[3,5,6].

SIRT7 is a nuclear enzyme both present in the nuclear lumen and the nucleolus. In the latter, it controls ribosomal DNA transcription[7,8] and maintains chromatin packing as protection from homologous recombination and damage[9,10]. Outside of the nucleolus, SIRT7 is responsible for gene promoter regulation and is especially relevant in cancer, where it is often overexpressed and helps to repress tumor suppressor genes and activate metastatic pathways[11–15]. These functions depend on the SIRT7-mediated deacetylation of nucleolar proteins and of the histone mark H3K18ac[11,16,17]. Accordingly, SIRT7 inhibitors have been proposed as potential epigenetic cancer chemotherapy[18].

SIRT7 is also a key regulator of H3K36ac in cells[15,19]. This modification lowers nucleosome stability[19] and favors euchromatin formation[20], and it is enriched at transcription start sites[20]. Importantly, H3K36ac competes with methylation of H3K36, a PTM that plays a role in many pathways including DNA damage-signaling[21], which may explain the role of SIRT7 in DNA repair[19]. Together, the H3K18 and H3K36 deacetylase activities of SIRT7 render it an important regulator of gene expression and chromatin integrity. However, little is known about the molecular mechanisms governing SIRT7

[1]Laboratory of Biophysical Chemistry of Macromolecules (LCBM), Institute of Chemical Sciences and Engineering (ISIC), School of Basic Sciences (SB), EPFL, Lausanne, Switzerland. [2]Laboratory of Biological Electron Microscopy (LBEM), Institute of Physics (IPHYS), School of Basic Sciences (SB), EPFL, Lausanne, Switzerland. [3]Department of Fundamental Microbiology (DMF), Faculty of Biology and Medicine (FBM), University of Lausanne, Lausanne, Switzerland. [4]These authors contributed equally: Carlos Moreno-Yruela, Babatunde E. Ekundayo. ✉e-mail: carlos.morenoyruela@epfl.ch; beat.fierz@epfl.ch

function, and recent studies underscore that its role in cancer is poorly understood and highly context-dependent, leading to both tumor promotion and tumor suppression[22–24].

SIRT7 activity shows characteristics unlike those of any other sirtuin. In vitro, SIRT7 presents both lysine deacylase and ADP-ribosylase activities[11,25–28], and it has preference for the long-chain acyl-lysine modification decanoyl-lysine (Kdec)[29]. Its deacylase activity on peptide and protein substrates requires binding to nucleic acids, in contrast with the rest of the sirtuin family[30,31]. SIRT7 is most active when bound to nucleosomes, showing affinity in the nanomolar range towards unmodified nucleosomes[32] and targeting H3K18ac and H3K36ac nucleosome substrates with unique selectivity[19,33,34]. The main structural differences of SIRT7, which may explain its behavior, are the two positively charged domains that flank its catalytic domain, proposed to mediate nucleic acid binding and activation[30,31,35,36]. A crystal structure of a part of the N-terminal domain shows helical folding reminiscent of DNA-binding domains[37], but no other structural evidence has been provided to date.

Here, we reveal the structural basis for SIRT7-specific substrate recognition and activation by nucleosomes. We synthesize nucleosomes with mechanism-based inhibitors at lysines 36 and 18 in H3 to trap the active conformation of SIRT7[38,39]. We then solve the structures of SIRT7 bound to a nucleosome and contacting H3K36 or H3K18 by cryogenic electron microscopy (cryo-EM) to a resolution of 2.8 and 3.5 Å, respectively. These structures reveal its atypical nucleosome-binding mode: we find that SIRT7 is recruited to the nucleosome by its extended N-terminal domain, which traverses the nucleosome surface and binds to both the acidic patch and nucleosomal DNA. At the same time, the catalytic domain engages both DNA gyres at the H3-tail exit site of the nucleosome. Strikingly, SIRT7 adopts different conformations depending on the target-lysine, H3K36 or H3K18, and the higher structural heterogeneity of the H3K18-bound complex explains the lower efficiency of SIRT7 towards this target compared to H3K36. Moreover, by comparing H3K18- and H3K36-bound structures, we find that distortion of the nucleosomal DNA structure is only needed for H3K18 targeting. Finally, structure-based mutagenesis affords SIRT7 variants with altered

substrate selectivity and unveil a key catalytic domain loop responsible for the H3K36-specific targeting of SIRT7 in vitro and in cells.

## Results

### Thioureas afford mechanism-based SIRT7:nucleosome complexes

SIRT7 is highly activated by nucleosome binding and is most active against H3K36 substrates, followed by H3K18[19,32]. To reveal the structural basis of how this enzyme can specifically target two distinct sites within a nucleosome, we decided to stabilize nucleosome-bound SIRT7 contacting H3K36 or H3K18. We envisioned the use of mechanism-based sirtuin inhibitor warheads at these two positions. Typical functional groups are thioureas[39,40], which substitute nicotinamide at the 1' ribose position of NAD+ and stall the sirtuin mechanism at one of the covalent intermediates[38,41], or form an alternative imine adduct, as recently found in a SIRT6:thiourea complex (Fig. 1a)[39]. To generate these constructs, we first introduced methylthiourea (MTU) and decylthiourea (DTU) modifications onto synthetic H3 peptides on resin[42]. Then, we followed with two- or three-fragment native chemical ligation (NCL) protocols to obtain full-length H3 modified at K18 or K36, respectively (Fig. 1b)[43].

With the full-length H3 constructs in hand, we reconstituted nucleosomes using the nucleosome-positioning 601 DNA sequence[44] with 20 bp flanking additions that promote SIRT7 activity at H3K36ac[19]. Thiourea-containing nucleosomes formed more stable complexes with SIRT7 than unmodified nucleosomes, as observed by electrophoretic mobility shift assay (EMSA). These complexes followed 1:1 as well as 2:1 stoichiometry, where both H3 molecules are engaged by SIRT7 (Fig. 1c and Supplementary Fig. 1B). Importantly, complex stabilization was NAD+-dependent, which demonstrated the formation of mechanism-based adducts at the thiourea position (Fig. 1a). Complexes linked at H3K36 and at H3K18 were stabilized with similar performance (Fig. 1d and Supplementary Fig. 1C), with both the shorter MTU and longer DTU functionalities that mimic the preferred Kac and Kdec substrates of SIRT7 (Fig. 1b)[29,45]. Thus, mechanism-based adduct formation does not depend on relative enzyme specificity.

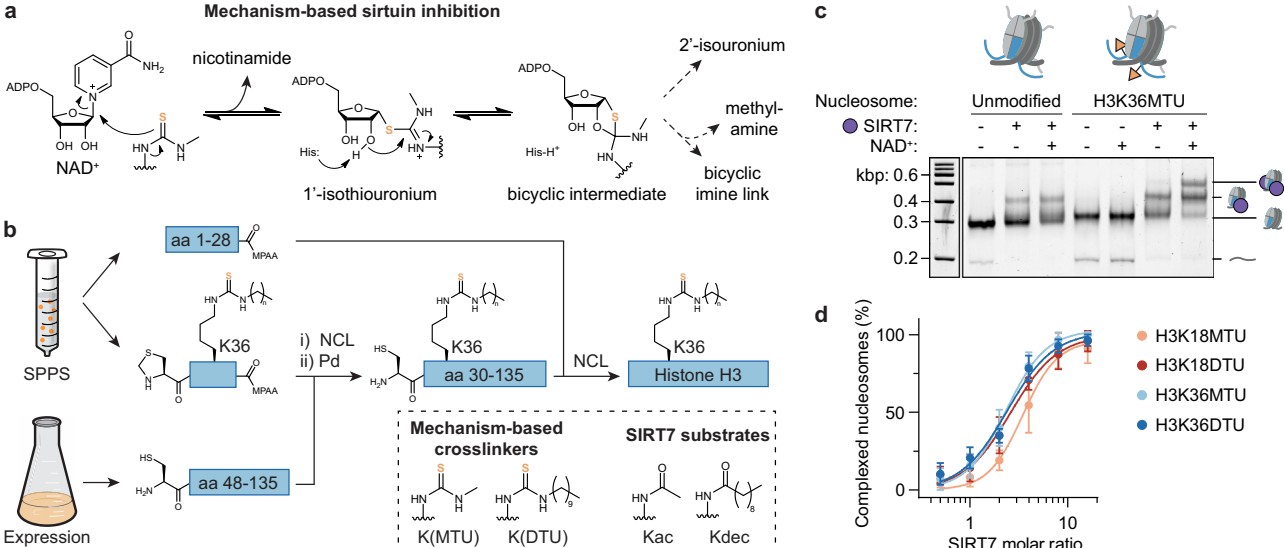

**Fig. 1 | Thiourea-modified histone H3 stabilizes the SIRT7-nucleosome interaction. a** Mechanism-based sirtuin inhibition by thioureas[38,39]. **b** Semi-synthetic approach to generate full-length H3 with methylthiourea (MTU), decylthiourea (DTU), acetyl (ac), or decyl (dec) modifications at K36 (see Supplementary Information−Chemical Synthesis for K18 modifications). Peptides covering amino acids (aa) 1–28 and 29–46 were prepared by solid phase peptide synthesis (SPPS) and linked to a recombinant fragment by native chemical ligation (NCL). MPAA:

4-mercaptophenylacetic acid. Overall H3 yield: 12–22% in 0.5–1.1 mg scale. See Supplementary Fig. 1A for sample HPLC and MS spectra. **c** SIRT7 EMSA of nucleosome samples with and without H3K36MTU modification (SIRT7-nucleosome molar ratio: 3). See Supplementary Fig. 1B for H3K18DTU EMSA. **d** Quantification of SIRT7-nucleosome complex formation with H3K18 or H3K36 thioureas. Error bars represent mean ± SD (*n* = 3) of distinct samples. See Supplementary Fig. 1C for representative EMSAs. Source data are provided as a Source Data file.

## SIRT7 binds to DNA and the acidic patch to target H3K36

We proceeded to analyze the complex of full-length SIRT7 with H3K36MTU-containing nucleosomes by single-particle cryo-EM (Supplementary Fig. 2A). Three-dimensional (3D) reconstruction to an overall resolution of 2.8 Å displayed SIRT7 bound to the nucleosome edge (Fig. 2a and Supplementary Figs. 3 and 4). The cryo-EM map revealed distinct densities for SIRT7 side chains, especially at the nucleosome-binding interface and for the mechanism-based adduct on the nucleosome, which enabled the detailed modeling of

a high-resolution structure of the complex (Fig. 2b and Supplementary Fig. 5).

In our model, the catalytic domain binds to the side of the nucleosome at the H3 tail exit site. The SIRT7 substrate pocket accommodates the H3K36 side chain securing it in place using residue F239. The map permitted reconstruction of the mechanism-based adenosine diphosphate ribose (ADPr)-MTU conjugate (Fig. 2c), and the observed density and LC-MS analysis were found to be consistent with the bicyclic intermediate also observed in the X-ray

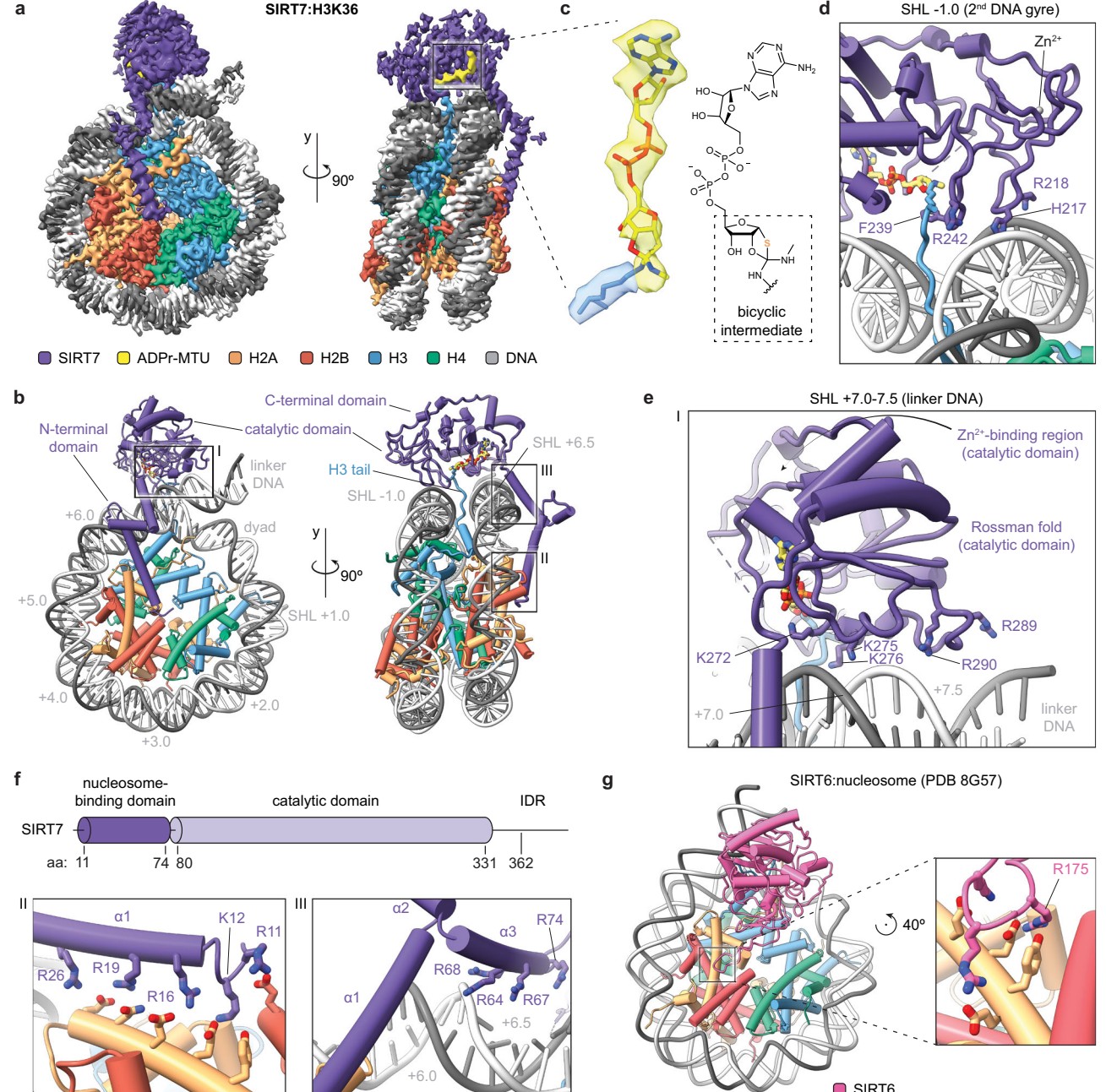

**Fig. 2 | Cryo-EM structure of SIRT7 bound to the H3K36MTU nucleosome.**
**a** Cryo-EM reconstruction showing SIRT7 bound to the nucleosome, with a mechanism-based cross-link to H3K36 (ADPr-MTU: adenosine diphosphate ribose-methylthiourea adduct). See Supplementary Figs. 3–4 for data processing workflow, resolution, and angular particle distribution. **b** Model of SIRT7(10-362) bound to the H3K36MTU nucleosome. See Supplementary Fig. 5 for model quality. The gray numbers indicate superhelical locations (SHL) along the nucleosomal DNA. For a detailed view of regions I–III, see **e**, **f**. **c** Cryo-EM density of the ADPr-MTU covalent adduct and fitting of a mechanism-based bicyclic intermediate model. See Supplementary Fig. 6 for LC-MS data. **d** Detailed interactions of the SIRT7 catalytic domain with the 2nd DNA gyre and the H3 tail. **e** Detailed interactions with linker DNA, corresponding to frame "I" in Fig. 2b. **f** SIRT7 domain organization (IDR: intrinsically disordered region) and detailed interactions of the N-terminal (nucleosome binding-) domain with DNA and histones. Panels correspond to frames "II" and "III" in Fig. 2b. **g** Cryo-EM structure of SIRT6 bound to a nucleosome, highlighting the arginine anchor (R175) interaction with the nucleosome acidic patch (PDB 8G57)[52].

structure of SIRT5 bound to a thioamide[46] (see also Supplementary Fig. 6). This was in contrast with a recent structure where ADPr in SIRT6 formed a bicyclic imine link with H3K9MTU due to elimination of methylamine[39] (Fig. 1a).

Contrary to enzymes that methylate[47] or demethylate[48] H3K36, SIRT7 does not unwrap DNA to target this residue from the histone octamer face. Instead, the SIRT7 catalytic domain presents two basic patches that bind in a cross-DNA gyre conformation (Fig. 2b), similar to PWWP H3K36me3 reader domains (Supplementary Fig. 7)[49]. In this configuration, multiple loops surrounding the SIRT7 catalytic pocket establish contacts with the backbone of both gyres of nucleosomal DNA as well as with extra-nucleosomal 'linker' DNA. Three residues away from F239, the side chain of R242 binds to DNA (3.1 Å distance) at the 2nd DNA gyre (superhelical location, SHL, −1.0). Here the DNA is also contacted by a neighboring loop within the $Zn^{2+}$-binding region of SIRT7 via residues H217 and R218 (3.6–4.6 Å, Fig. 2D). Opposite of F239, a highly conserved SIRT7 loop[36] within the $NAD^+$-binding Rossmann fold makes weak electrostatic interactions (4.7–6.2 Å) with DNA at SHL + 7.0, using lysines 272, 275, and 276 (Fig. 2e). The Rossman fold further extends along the linker DNA and reaches the phosphate backbone at SHL + 7.5 with R289 (3.8 Å) and, to a lesser extent, R290 (6.2 Å).

Remarkably, SIRT7 interacts with the octamer face and the H2A–H2B acidic patch via its N-terminal domain, which folds into three α-helices that extend across the nucleosome (Fig. 2b). Helix α1 binds to the classical acidic patch residues E105 of H2B with R11, and E61, D90, and E92 of H2A with K12 (Fig. 2f). The SIRT7 α1 helix then follows the H2A helix and further contacts H2A E64, N68 and D72 with arginines 16, 19 and 26 (Fig. 2f), and it forms weak electrostatic interactions (5.4–6.1 Å) with arginines 18 and 21. The end of helix α1 sits close to SHL + 6.0 and features three additional arginines, 30, 34, and 37, of which only R37 makes direct contact (4.9 Å) with the DNA phosphate backbone. Helix α2 serves as the 'elbow' of the N-terminal domain and lets helix α3 accommodate in the major groove at SHL + 6.5, where it contacts DNA with arginines 64, 68, and 74 (2.8–3.3 Å, Fig. 2f). Thus, the SIRT7 N-terminal domain serves as a multivalent nucleosome-binding domain through eight side chains forming strong electrostatic interactions, among other contacts. This atypical binding mode explains how SIRT7 can target preferentially H3K36ac PTMs on the side of the nucleosome while interacting with the acidic patch[32].

Multiple nucleosome-modifying enzymes interact with the acidic patch for chromatin binding, often through a single arginine anchor[50,51]. The closely related sirtuin SIRT6 shares such binding mode thanks to a specific loop insertion within the catalytic domain[39,52] (Fig. 2g). In contrast, according to our structure the SIRT7 catalytic domain sits much further from the center of the nucleosome, and the N-terminal domain interacts with the same anchor hotspot via an extended set of basic residues from R11 to R26, exhibiting non-canonical acidic patch interactions unlike any other enzyme shown to date (Fig. 2f)[51].

Finally, three limited regions are missing in our SIRT7 model: the outermost N-terminus (aa 1–9), a flexible loop within the catalytic domain (aa 120–138), and part of the C-terminal domain (aa 363–400). The loop around aa 120–138 is predicted by AlphaFold 3[53] to contact DNA close to the nucleosome-binding domain (Supplementary Fig. 8), albeit at lower confidence than the rest of the catalytic domain. In agreement with this prediction, our cryo-EM map shows low-resolution density within the corresponding region, which suggests that this loop is disordered and may contribute to nucleosome binding. As for the C-terminal tail, this fragment is predicted to be an intrinsically disordered region (IDR, Fig. 2f) containing a nuclear-localization sequence[36]. The C-terminal tail is also predicted to bind to nucleic acids and has been shown to mediate activation by DNA[30]. In our model, a part of it (aa 331–362) is visible contacting the top of the

catalytic domain (Fig. 2b). The remaining IDR may further project into the solution, allowing nucleic acid interactions in the vicinity.

## Targeting H3K18 involves SIRT7 and DNA structural changes

To evaluate how SIRT7 contacts its substrates at H3K18 and to compare target-lysine specific binding modes, we next reconstituted a SIRT7 complex with a H3K18DTU-modified nucleosome for analysis by cryo-EM (Supplementary Fig. 2B). However, while the mechanism-based cross-link performed equally well as compared to the H3K36MTU (Fig. 1d), the SIRT7:H3K18DTU-nucleosome complex required further stabilization by gradient fixation[54] in order to obtain density in the cryo-EM maps. This suggests that SIRT7 binding is more dynamic when bound to H3K18 than to H3K36. Nonetheless, we obtained a cryo-EM map of the SIRT7: H3K18DTU complex at a medium resolution (3–7.5 Å, and 3.5 Å overall, Fig. 3a and Supplementary Figs. 9–10), which enabled modeling of the SIRT7 domains into the cryo-EM map (Fig. 3b and Supplementary Fig. 11A).

The N-terminal and catalytic domains of SIRT7 interact with similar nucleosome regions in both the H3K18-bound and H3K36-bound models, though they exhibit key differences in orientation that emphasize a substrate-specific conformation (Fig. 3c). Notably, the catalytic domain adopts a distinct conformation when bound to H3K18, indicating that movement within this domain is necessary for H3K18 targeting (Fig. 3c). In concert with this reorientation of SIRT7, the linker DNA contacting the catalytic domain is bent inwards towards the face of the nucleosome by 6° relative to its position in the H3K36-bound structure (Supplementary Fig. 11B). This shift of the linker DNA allows contact with the H2A C-terminal tail that has also already been observed in histone H1-induced linker DNA bending (Supplementary Fig. 11C)[55–57]. Moreover, R289 forms a closer interaction with the linker DNA region, whereas other residues contacting both linker DNA (K272, 275 and 276, and R290) and nucleosomal DNA (H217, R218, and R242), are positioned further away from the DNA backbone (>7 Å, Fig. 3d). These changes in distance reveal weakened electrostatic interactions at the SIRT7-DNA interface when targeting H3K18, possibly accounting for the less defined cryo-EM density and indicating increased flexibility compared to the H3K36-bound map.

Together with the overall reduced resolution of the SIRT7 catalytic domain in comparison to the rest of the cryo-EM map, we found less defined density in the substrate-binding pocket. In consequence, we did not model the H3 tail or the ADPr-DTU cross-link (Fig. 3a). Since the distance between the substrate-locking F239 residue and the H3 tail exit site is similar in this model compared to the H3K36-bound structure, the H3 tail must form a loop along the linker DNA to place the H3K18 side chain in the SIRT7 active pocket. In agreement, we observed additional putative H3 density within the open space generated by the 6° displacement of the linker DNA (Supplementary Fig. 11c).

The shift of the catalytic domain further alters the overall conformation of the nucleosome-binding domain and, in particular, changes the bending angle of helix α2 by 30° (Fig. 3c), which in this case is more similar to the published partial crystal structure of the SIRT7 N-terminus[37]. Importantly, both helices α1 and α3 maintain contact with the acidic patch and the DNA compared to the previous conformation, which suggests that the positioning of the nucleosome-binding domain is primarily mediated by these key interactions and not by the location of the PTM on the nucleosome. In contrast, the hinge formed around helix α2 generates the necessary flexibility for the enzyme to shift pose.

The overall comparison of the two cryo-EM datasets indicates that SIRT7 adopts a more defined conformation for H3K36 than for H3K18 binding, the latter of which also includes a distortion of nucleosomal DNA conformation. To test how these different binding modes relate to the enzymatic activity of SIRT7, we performed deacetylation assays. LC-MS analysis of the products revealed higher

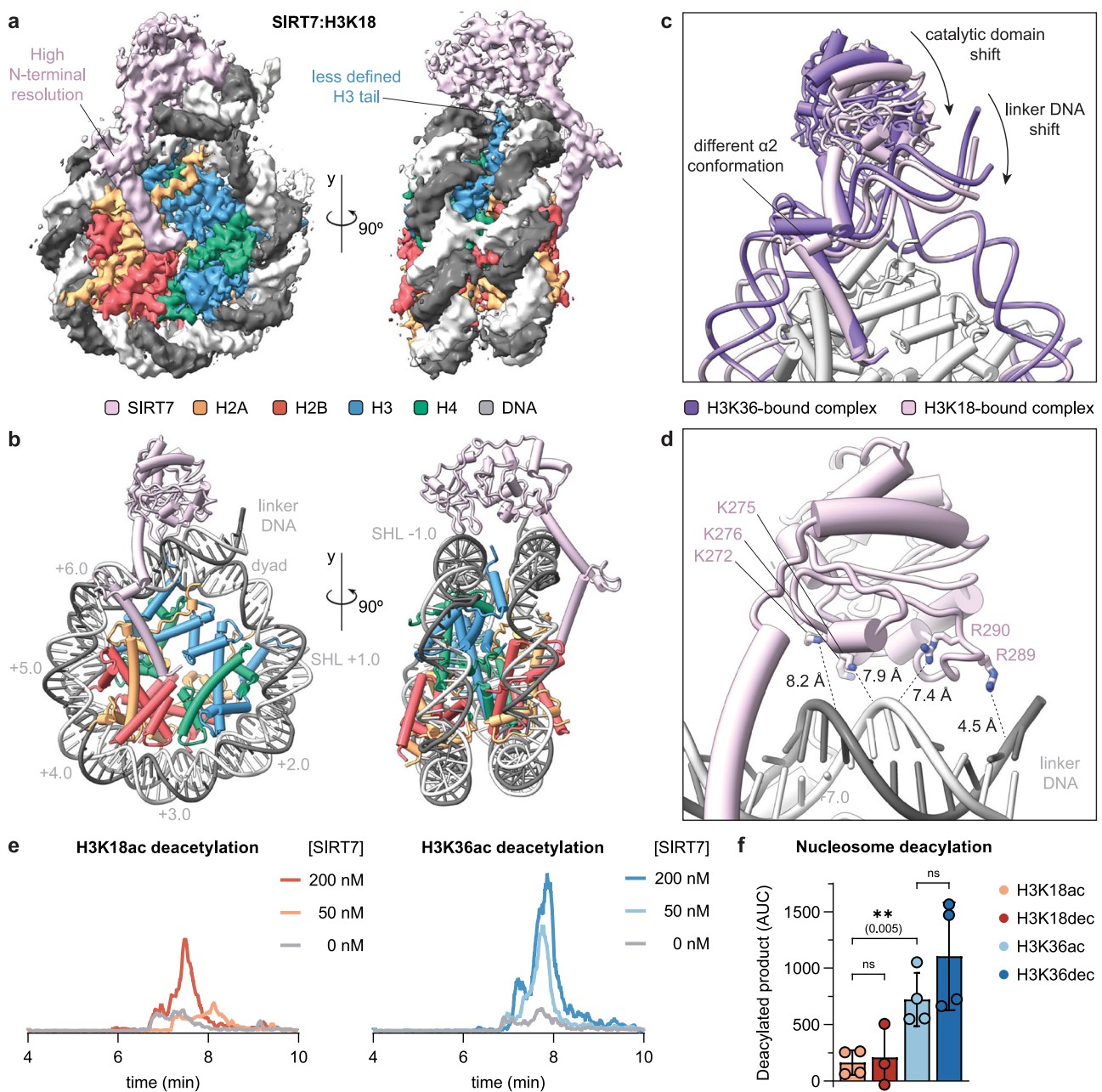

**Fig. 3 | Altered binding pose of SIRT7 in complex with the H3K18DTU nucleosome. a** Cryo-EM reconstruction showing SIRT7 bound to the H3K18DTU nucleosome. See Supplementary Figs. 9–10 for data processing workflow, resolution and angular particle distribution. **b** Model of SIRT7(10-362) bound to the H3K18DTU nucleosome. See Supplementary Fig. 11 for model quality. SHL: superhelical location. **c** Comparison of SIRT7:nucleosome complexes bound to the H3K36 and H3K18 positions, with highlighted key differences. **d** Distance between residues that contact DNA for H3K36 binding and the DNA backbone in the H3K18-bound conformation. **e** LC-MS analysis of nucleosome deacetylation by SIRT7, showing the extracted ion chromatogram of the corresponding deacetylated product ($[M + 21H]^{21+}$ ion). Nucleosome concentration was 200 nM. **f** LC-MS quantification of nucleosome deacylation by SIRT7 (50 nM concentration). Data represent the area under the curve (AUC, $[M + 21H]^{21+}$ ion) of extracted ion chromatograms minus background. Error bars represent mean ± SD ($n = 3$ or 4) of distinct samples. Statistical analysis: unpaired $t$ tests, two-tailed, ns: $p > 0.05$. Source data are provided as a Source Data file.

SIRT7 activity on H3K36ac compared to H3K18ac substrates (>4-fold higher activity at 50 nM SIRT7 concentration, Fig. 3e), in line with previous reports[19,33]. Furthermore, although SIRT7 shows greater activity towards long acyl modifications in peptide substrates[29], we did not observe significant differences between Kac and Kdec substrates at either H3K36 or H3K18 within nucleosomes (Fig. 3f). In summary, our results indicate that nucleosome binding directs SIRT7 activity preferably towards deacetylation of H3K36, and that access to the H3K18 substrate is disfavored, as it requires the adoption of a more dynamic conformation.

## The nucleosome-binding domain recruits SIRT7 to chromatin
Using the two structures for H3K36 and H3K18-contacting SIRT7 as a basis, we investigated how the observed interactions of the N-terminal nucleosome-binding domain with the histone octamer and, in particular, the acidic patch drive overall binding and activity. To this end, we generated two new SIRT7 variants. In the first, we introduced two mutations into the nucleosome-binding domain, targeting the arginine anchor that directly contacts the acidic patch (SIRT7 R11A, K12A). In a second variant, we removed the complete helix α1 of the nucleosome-binding domain (SIRT7 40–400) (Fig. 2f). While SIRT7(R11A, K12A)

showed similar binding to nucleosomes compared to the wild-type enzyme, the N-terminal truncation in SIRT7(40–400) reduced binding significantly (Fig. 4a, b and Supplementary Fig. 12A). Therefore, the interaction of SIRT7 with the nucleosome relies on multivalent engagement of the histone octamer beyond the acidic patch. In agreement, multiple arginine side chains along the SIRT7:octamer interface exhibit highly defined cryo-EM densities (Supplementary Figs. 5 and 11). Interestingly, this contrasts with the common arginine anchor-based binding modes observed in other nucleosome-modifying enzymes[51].

We then tested the activity of these mutants, as well as an active site mutant (H187Y)[11,58], towards H3K18ac and H3K36ac in reconstituted nucleosomes in vitro. For these assays, we chose SIRT7 concentrations resulting in comparable activity towards H3K36 and H3K18 (3 and 50 nM, respectively, Fig. 4c). As expected, the active site mutant H187Y did not exhibit much activity on either substrate (Fig. 4d)[11,58]. Conversely, the activity of SIRT7(R11A, K12A) was increased towards H3K18ac deacetylation (Fig. 4d). This finding highlights that SIRT7 does not rely on an arginine-anchor mechanism for maintaining its ability to deacetylate nucleosomes under these conditions. In contrast, mutating arginine-anchor residues in SIRT6 resulted in a loss of enzymatic activity[39,52], further emphasizing the differences in substrate recognition among the two class IV sirtuins. Interestingly, also the truncated SIRT7(40–400) construct showed activity similar to the wild-type enzyme (Fig. 4d). This can be rationalized since, under our assay conditions, dissociation from the product is the rate-limiting step for SIRT7 mononucleosome deacetylation[32]. Thus, faster substrate release of SIRT7(40–400), due to its lower binding affinity, could be a plausible explanation for the increased deacetylation kinetics.

Based on these results, we hypothesized that the role of the SIRT7 N-terminal domain lies in its specific recruitment to nucleosomes over other nucleic acids. We thus challenged SIRT7 with increasing amounts of competitor DNA, while monitoring its deacetylation activity on nucleosomes. Indeed, SIRT7(40–400) was efficiently outcompeted by free DNA, while the wild-type enzyme retained its activity independent of added competitor DNA (Fig. 4e). Next, we transfected a stable HEK293F *SIRT7*⁻/⁻ cell line[27] with SIRT7 wild type or SIRT7(40–400) and found that the wild type induced lower acetylation levels than the truncated construct, especially against H3K18ac marks (Fig. 4f and Supplementary Fig. 13). This was consistent with SIRT7(40–400) being a less efficient deacetylase within the cellular environment. Thus, the full nucleosome-binding domain is necessary for efficient chromatin recruitment of SIRT7 in a complex environment.

### SIRT7 site selectivity relies on specific contacts with DNA

Careful comparison of the catalytic domain conformation between the H3K36- and H3K18-bound structures revealed marked conformational changes of DNA-binding loops (Fig. 3c). To analyze the contribution of each loop to the substrate selectivity of SIRT7, we mutated key positively charged amino acids to alanine in each of them separately (loop 1: H217A, R218A; loop 2: K272A, K275A, K276A; and loop 3: R289A, R290A, Fig. 5a). Overall, the three loop-mutants showed activity on both H3K18ac and H3K36ac substrates, although with interesting differences in site selectivity (Fig. 5b).

When replacing H217 and R218 with alanine within loop 1, SIRT7 activity towards H3K18ac was increased (Fig. 5c). This indicates that binding to the 2nd DNA gyre on the nucleosome, thereby positioning SIRT7 over the H3 tail exit site, is suboptimal for K18 deacetylation. Releasing the enzyme from these constraints may generate a more dynamic complex with relaxed specificity.

Strikingly, mutating lysines K272, K275, and K276 in loop 2 to alanine resulted in even higher SIRT7 deacetylation activity on H3K18ac, leading to the inversion of substrate preference towards H3K18 over H3K36 (Fig. 5d). Indeed, when transfecting HEK293F *SIRT7*⁻/⁻ cells with this mutant, the nuclear H3K18ac levels decreased to

a much larger extent than with wild type SIRT7, while H3K36ac remained at a similar level (Fig. 5e and Supplementary Fig. 13), which confirmed that SIRT7 (K272A, K275A, K276A) is more active against the former modification and the overall inversion of SIRT7 substrate selectivity within the cellular environment. This mutant thus reveals that the loop between K272 and K276 is key for the substrate specificity of SIRT7. Based on our structures, we argue that binding of this loop to DNA forces the Kac substrate pocket to face the H3 tail exit site, where H3K36 is located, and thus makes it difficult for it to target other substrates in the H3 tail. To further test this hypothesis, we measured SIRT7 activity on H3K18ac nucleosomes lacking linker DNA (beyond SHL + 7.0) and found an increase in wild-type activity in line with reduced specificity (Supplementary Fig. 12C).

Finally, SIRT7 containing point mutants R289A, R290A within loop 3 showed lower activity, in particular on H3K36ac nucleosomes (Fig. 5c). Thus, linker DNA interactions are important for H3K36 targeting and contribute to the substrate selectivity.

In summary, the extended N-terminal domain of SIRT7 is crucial for recruitment to chromatin within the nuclear environment and its activity on H3 substrates, positioning the catalytic domain to tightly interact with both DNA gyres by the exit of the N-terminal H3 tail. The DNA contacts of the catalytic domain have evolved towards optimal H3K36 deacetylation and confer selectivity against other substrates. Conversely, the relaxation of DNA interactions increases SIRT7 dynamics and allows the enzyme to target H3K18. Loosening of the complex structure via mutations targeting the SIRT7:nucleosome interface further promotes H3K18 deacylation activity both in vitro and in cells. Together, the DNA targeting loops, in particular, residues K272, K275, and K276, ensure H3K36ac selectivity, highlighting a new chromatin targeting mechanism.

## Discussion

SIRT7 is a driver of tumor growth and metastasis in various cancers and has been proposed as therapeutic target[11,14,15,24]. However, its complicated biochemistry, in particular the dependence on nucleosomal substrates or activation by nucleic acids, as well as the lack of structural information, has impeded the study and inhibitor development of this histone modifier.

Here, we have determined the cryo-EM structures of SIRT7 in complex with the nucleosome, targeting two key substrate residues, H3K36 and H3K18. As in other recent studies[39,48], the use of mechanism-based inhibitor groups was crucial to stabilizing the active conformation of SIRT7 contacting specific target residues[34], allowing us to identify the mechanisms of substrate selectivity in an unprecedented manner. As mechanism-based cross-linking stabilizes substrate complexes regardless of their relative conversion rates[59,60], we envision that this strategy will be widely applicable to a variety of enzyme-substrate complexes and will greatly advance the study of their mechanism of action.

Both members of class IV sirtuins, SIRT6 and SIRT7, bind to nucleosomes with nanomolar affinity and are activated by these interactions[32,61]. In the case of SIRT6, the mechanism of activation is proposed to be through multivalent interaction of its disordered C-terminal domain with DNA[62], while the catalytic domain binds to the acidic patch with a SIRT6-specific arginine anchor crucial for its activity[39,52]. In contrast, here we show that SIRT7 does not rely on a single anchor, but that it establishes a multivalent interaction surface across the nucleosome via the SIRT7-specific nucleosome-binding domain. This domain is especially important for recruitment within the nucleus, as lack of contact with the histone octamer makes SIRT7 sensitive to DNA inhibition.

While the nucleosome-binding domain forms interactions across the face of the nucleosome, the catalytic domain contacts both DNA gyres on the side of the nucleosome. This is unprecedented in chromatin-associated enzymes and key for SIRT7 selectivity towards

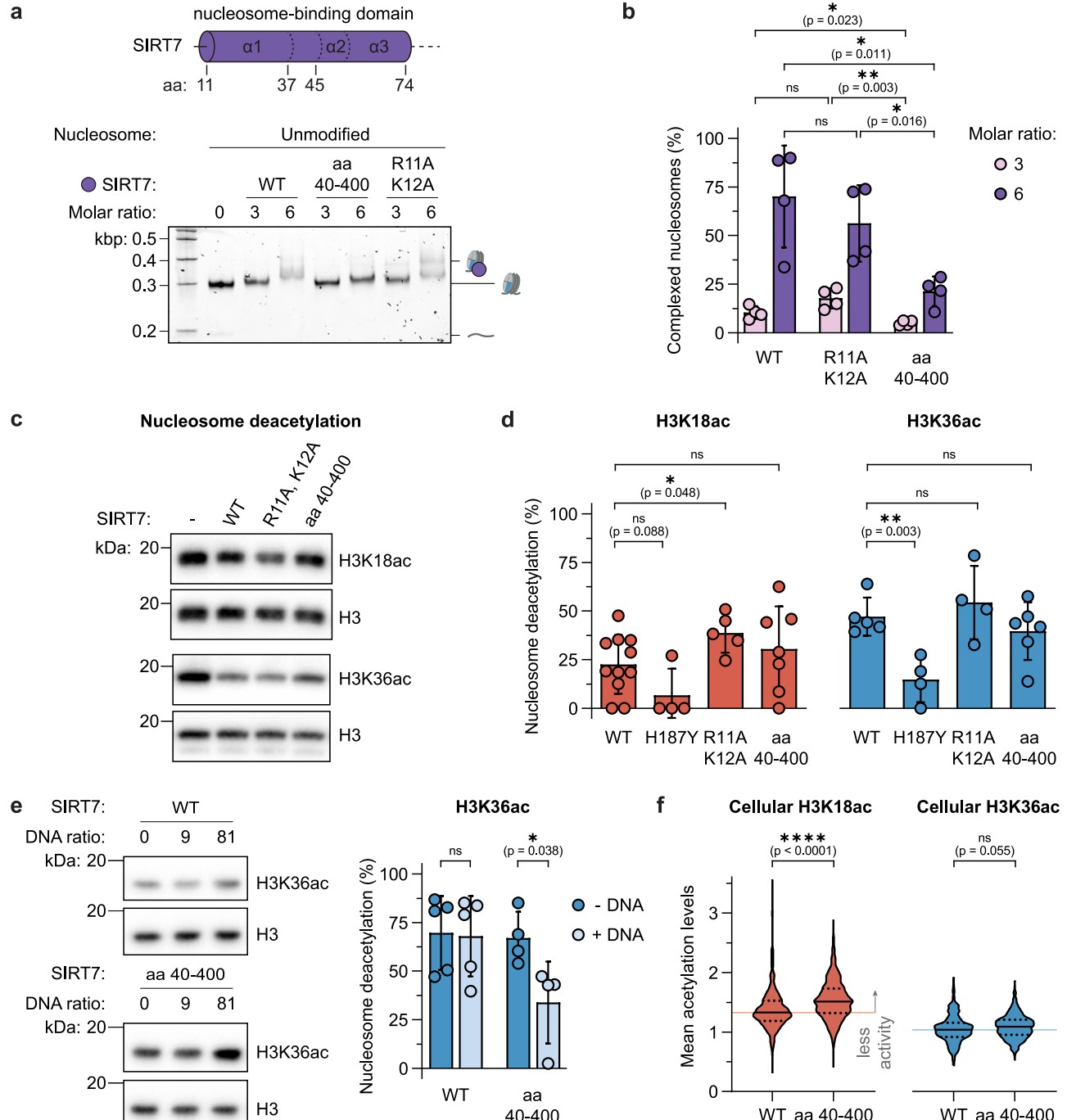

**Fig. 4 | Role of the N-terminal domain in nucleosome binding and deacetylation. a** EMSA of unmodified nucleosomes with wild-type SIRT7 and nucleosome-binding domain mutants. See Supplementary Fig. 12A for replicate gels. **b** Quantification of SIRT7:nucleosome complex formation with unmodified nucleosomes and mutants as indicated. Error bars represent mean ± SD ($n = 4$) of distinct samples. Statistical analysis: unpaired $t$ tests, two-tailed, ns: $p > 0.05$. **c** SIRT7 wild type and mutant activity on H3K18ac and H3K36ac nucleosomes measured by western blot. SIRT7 concentration was 50 nM and 3 nM, respectively. **d** Quantification of nucleosome deacetylation normalized to H3 loading, at 50 nM SIRT7 for H3K18ac and 3 nM SIRT7 for H3K36ac. Error bars represent mean ± SD ($n = 4$–11, as shown) of distinct samples. Statistical analysis: unpaired $t$ tests, two-tailed, ns: $p > 0.05$. **e** Inhibitory effect of free DNA (187 bp) on SIRT7 activity, measured by western blot,

and quantification of the effect of 81 equivalents of DNA, normalized to H3 loading. See Supplementary Fig. 12B for complete titration. Error bars represent mean ± SD ($n = 4$ or 5) of distinct samples. Statistical analysis: unpaired $t$ tests, two-tailed, ns: $p > 0.05$. **f** Violin plot (truncated) of H3K18ac and H3K36ac levels in HEK293F $SIRT7^{-/-}$ cells[27] upon transient transfection with SIRT7-mCherry constructs as indicated, measured by immunofluorescence. Internal lines represent median and quartile values of $n \geq 4$ distinct experiments, and light background line indicates median value for WT SIRT7. Statistical analysis: Kruskal–Wallis and Dunn's multiple comparisons tests, non-parametric two-tailed, ns: $p > 0.05$, $p(\text{H3K18ac}) = 2 \cdot 10^{-11}$. See Supplementary Fig. 13 for SIRT7 western blots, representative microscopy images, and control H4K16ac quantification. Source data are provided as a Source Data file.

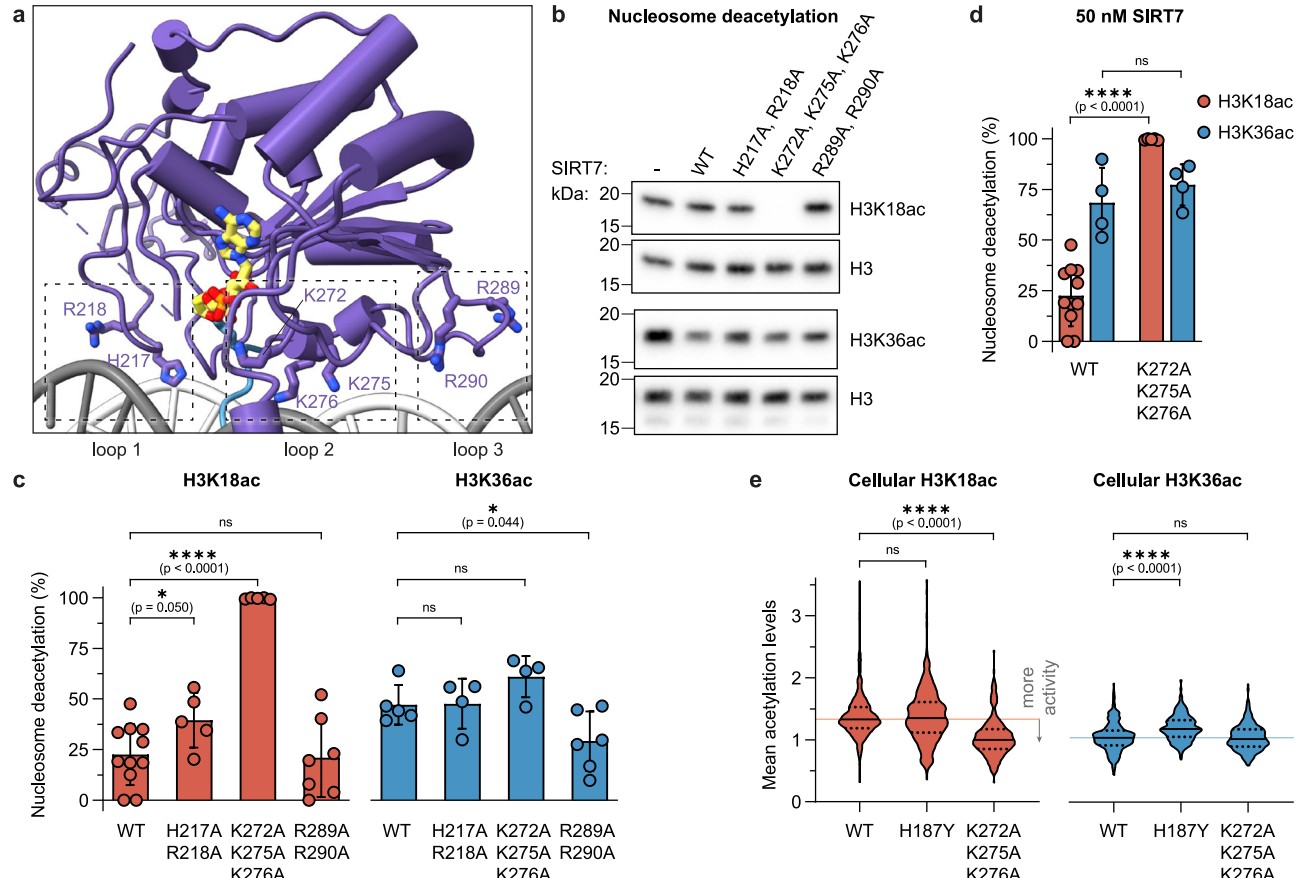

**Fig. 5 | Loops flanking SIRT7 catalytic domain control substrate selectivity.**
**a** Summary of mutated catalytic domain loops. **b** SIRT7 wild type and mutant activity on H3K18ac and H3K36ac nucleosomes measured by western blot. SIRT7 concentration was 50 nM and 3 nM, respectively. **c** Quantification of nucleosome deacetylation normalized to H3 loading, at 50 nM SIRT7 for H3K18ac (top) and 3 nM SIRT7 for H3K36ac (bottom). See Supplementary Fig. 12D for data on helix α3 mutant K72A, R73A, R74A. Error bars represent mean ± SD ($n = 4$–11, as shown) of distinct samples. Statistical analysis: unpaired $t$ tests, two-tailed, ns: $p > 0.05$, $p$(K272A, K275A, K276A) = 2 · $10^{-8}$. **d** Quantification of wild type and K272A, K275A, K276A mutant activity at 50 nM SIRT7 concentration. Error bars represent

mean ± SD ($n = 4$–11, as shown) of distinct samples. Statistical analysis: unpaired $t$ tests, two-tailed, ns: $p > 0.05$, $p$(K272A, K275A, K276A) = $2 \times 10^{-8}$. **e** Violin plot (truncated) of H3K18ac and H3K36ac levels in HEK293F $SIRT7^{-/-}$ cells[27] upon transient transfection with SIRT7-mCherry constructs as indicated, measured by immunofluorescence. Internal lines represent median and quartile values of $n \geq 3$ distinct experiments, and a light background line indicates median value for WT SIRT7. Statistical analysis: Kruskal–Wallis and Dunn's multiple comparisons tests, non-parametric two-tailed, ns: $p > 0.05$. See Supplementary Fig. 13 for SIRT7 western blots, representative microscopy images, and control H4K16ac quantification. Source data are provided as a Source Data file.

H3K36ac substrates. Since most of the DNA contacts are made by flexible loops within the $Zn^{2+}$-binding part of the catalytic domain[38] and, potentially, by an unstructured region within the Rossman fold (as predicted by AlphaFold 3), we hypothesize that SIRT7 is activated by an overall conformational rearrangement of the catalytic domain when bound to the nucleosome.

While this study focuses on SIRT7 activity on nucleosomes, earlier studies have indicated that its activity can be stimulated by free nucleic acids, in particular by 5S RNA through multiple interactions with the N-terminal domain[31]. Moreover, the disordered C-terminal domain of SIRT7 has also been predicted to bind to DNA and promote activation[30]. Key residues previously identified for enzyme activation by free RNA have, however, limited effect on nucleosome deacylation[31]. Thus, nucleosome- and RNA-dependent activation mechanisms seem to be distinct for SIRT7.

Both SIRT6 and SIRT7 target the H3 tail while remaining in contact with the nucleosome acidic patch. Their nucleosome-bound states remain dynamic—the enzymes sample a conformational landscape, allowing them to reach distinct substrate sites thereby conferring each enzyme with a distinct substrate selectivity[63]. Here, we capture SIRT7 in both H3K18- and H3K36-targeting conformations, corresponding to two functional states within this conformational landscape. To contact

H3K36, previous structures revealed that enzymes often require significant DNA unwrapping from the nucleosome surface[47,48]. In contrast, SIRT7 binds to nucleosomes tightly just above the H3-tail exit site, with the H3 tail directly entering the active site. At the same time, SIRT7 maintains extensive contacts across both nucleosomal DNA gyres and keeps the emerging linker DNA in place, forming a 'closed' conformation. In contrast, when targeting H3K18, SIRT7 has to partially disengage from the nucleosome to a more 'open' conformation, to allow the additional amino acids 19–35 of the H3 tail to loop under the bound enzyme to access the active site. This second conformation proved to be less defined than the previous H3K36-bound state and required additional cross-linking by gradient fixation for cryo-EM, which might indicate that the putative H3K18-bound conformational landscape is even broader. In this configuration, SIRT7:DNA contacts are relaxed, and the exiting linker DNA is bent, possibly due to the increased space requirement of the H3 tail and interactions with the reoriented catalytic domain. These conformational differences to the canonical nucleosome structure and the increase in structural heterogeneity of the complex provide a mechanistic explanation for the lower activity of SIRT7 towards H3K18 deacylation.

SIRT7 H3K36 targeting thus relies on DNA contacts via multiple loops within the catalytic domain, highlighting that multivalent

binding of SIRT7 to nucleosomes is optimized for H3K36 deacetylation versus other substrates. Instead of the traditional sequence-based enzyme selectivity filter, it appears that SIRT7 is selective based on its locked nucleosome-bound conformation and the distance to potential substrates. Moreover, we hypothesize that mutations in the DNA-contacting loops shift the conformational equilibrium from a 'closed' to a more 'open' state, which in turn favors enzymatic activity towards H3K18. This is indeed what we find when reengineering the enzyme. Importantly, we identified a key loop between K272 and K276 that determines SIRT7 substrate specificity in vitro and in cells, which will be important for future biochemical studies and tool development. Other reports have found that SIRT7 targets succinyl- and glutaryl-lysine PTMs within the nucleosome core (H3K122suc and H4K91glu, respectively[25,26]), which would require conformations different from those presented here. Thus, future research would determine whether the same interactions are involved in nucleosome core targeting and the required changes in chromatin structure.

The SIRT7 isozyme is present in all eukaryotes, but it only features a long helical N-terminal domain in insects and in vertebrates except for reptiles[36]. For example, in *D. melanogaster*, both N- and C-terminal tails are much longer than in humans but with a shorter N-terminal helical portion, and the "His-Arg" motif that interacts with the 2nd DNA gyre is absent[36]. Based on our data, this may indicate that SIRT7 has distinct PTM selectivity in different species. Further studies will thus be necessary to investigate the role of H3K36 deacetylation throughout evolution and across species.

In conclusion, we present the structural basis of SIRT7-nucleosome deacetylation at its preferred biological substrates H3K18ac and H3K36ac, and identify the helical N-terminal fragment as nucleosome-binding domain and key loops within the catalytic domain as drivers of selectivity towards H3K36ac. Moreover, the structures reveal that distinct conformations of the enzyme active site on the nucleosome, as well as their respective dynamics, drive substrate specificity within the histone tails, as opposed to the recognition of the peptide sequence context. These features are unprecedented among chromatin-binding proteins and represent a new paradigm in specific PTM targeting. The strategy used here, based on chemical modification of proteins using mechanism-based inhibitors[34], was essential to this work. We anticipate that such approaches will further advance the study of enzyme target selection and regulation, unveiling the mechanisms behind their functional diversity.

## Methods

Complete chemical synthesis and histone purification methods can be found in the Supplementary Information.

Fast protein liquid chromatography (FPLC) was performed on a ÄKTA Pure purification system equipped with UV detector and with columns as follow: HisTrap HP (5 mL, Cytiva) or HiFliQ (5 mL, Protein-nArk) for Ni-NTA immobilized metal affinity chromatography (IMAC), HiTrap SP HP (5 mL, Cytiva) for ion exchange chromatography (IEX), Superdex S200 Increase 5/150 GL for analytical size-exclusion chromatography (SEC, Cytiva), Superdex S200 10/300 GL (Cytiva) for octamer SEC, and Superose 6 Increase 10/300 GL (Cytiva) for SIRT7:nucleosome SEC. DNA and protein concentration was determined by UV-Vis spectroscopy, using absorption at 260 nm or 280 nm, respectively, as measured on a NanoDrop 2000 (ThermoFisher) by sample drop or a Lambda 365+ (PerkinElmer) instrument with quartz cuvettes. Protein high-resolution MS analysis was performed on a Waters Xevo G2-XS instrument equipped with a UV diode array and quadrupole-time-of-flight analysis systems. Histone octamers, nucleosomes, and SIRT7 mutants were analyzed on this system by HPLC-MS with a Waters Acquity BEH C4 column (1.7 μm, 100 × 2.10 mm, 300 Å) and gradients of eluent V (0.1% HCOOH in H2O) and eluent VI (0.1% HCOOH in MeCN) at a flow rate of 1 mL/min. Gels and western blot membranes were imaged using a ChemiDoc MP imaging system (Bio-Rad).

### Expression and purification of SIRT7 constructs

SIRT7 expression and purification were optimized from the protocol reported by Bolding et al.[29]. *E. coli* BL21 (DE3) cells were transformed with a pET100_D_TOPO plasmid encoding for SIRT7 or mutants thereof (generated by site-directed mutagenesis with the Q5 polymerase, New England Biolabs, cat. # M0491, followed by KLD enzyme mix treatment, New England Biolabs cat. # M0554), with an N-terminal fusion to a 6x histidine (6H) tag and a linker including a tobacco etch virus (TEV) protease cleavage site. Cells were cultured in 2–6 L autoinduction terrific broth (Formedium), or in lysogeny broth (Miller's LB broth, Condalab) supplemented for autoinduction (6.8 g/L KH2PO4, 7.1 g/L Na2HPO4, 3.3 g/L (NH4)2SO4, 5 g/L glycerol, 0.5 g/L dextrose, 2 g/L α-lactose), and with 100 μg/mL carbenicillin or ampicillin, at 37 °C with shaking at 200 rpm until OD600 = 0.6–0.8. Then, flasks were transferred to 18 °C and shaken at 150 rpm overnight. Cell pellets were obtained by centrifugation (4000 × g, 20 min, 4 °C), suspended in SIRT7 lysis buffer (20 mM Tris-HCl, 50 mM Na2HPO4, 700 mM NaCl, 1 mM EDTA, 2 mM DTT, 10% v/v glycerol, pH 7.5) with protease inhibitors (cOmplete EDTA free, Merck, cat. # 11697498001) and SerM nuclease A periplasmic fraction (only used for wild-type SIRT7), and flash-frozen for storage at −70 °C. Cells were lysed by sonication (10 s on and 30 s off, 60 s total time on) or by French press (3 × 15,000 bar, only done for wild-type SIRT7) and centrifuged (20,000 × g, 20–30 min, 4 °C). The supernatants were supplemented with imidazole to a concentration of 25 mM and purified by Ni-NTA FPLC using 5 → 70% gradients of buffer A (20 mM Tris-HCl, 50 mM Na2HPO4, 700 mM NaCl, 1 mM EDTA, pH 7.5) and buffer B (20 mM Tris-HCl, 50 mM Na2HPO4, 700 mM NaCl, 1 mM EDTA, 500 mM imidazole, pH 7.5). Fractions containing 6xH-SIRT7 were identified by SDS-PAGE (12% acrylamide gel) and combined, supplemented with DTT (2 mM) and TEV protease (cat. # Z03030, GenScript), and dialyzed against buffer C (50 mM K2HPO4, 50 mM NaCl, 1 mM DTT, pH 7.5) at 4 °C overnight. The incubation could be done without dialysis, in which case the resulting mixture was diluted 3–5 fold with buffer C before IEX FPLC. The mixture was then purified by IEX FPLC using 0→70% gradients of buffer C and buffer D (50 mM K2HPO4, 1000 mM NaCl, 1 mM DTT, pH 7.5), and fractions containing SIRT7 were pooled, concentrated using 4 mL and 0.5 mL 30 kDa MW cutoff centrifugal filters (Amicon Ultra, Merck Millipore) to concentrations >1 μM. The solutions were supplemented with glycerol (ThermoFisher, cat. # 327255000) to 10% wt/wt concentration, aliquoted, and flash-frozen for long-term storage at −70 °C. The final samples were analyzed by UV-Vis spectroscopy (calculated extinction coefficient at 280 nm: 53400 M⁻¹ cm⁻¹, used for all SIRT7 constructs) and SDS-PAGE to verify purity (Supplementary Fig. 17), and by HPLC-MS to verify mutations (Supplementary Figs. 22–23 and Supplementary Table 1).

All construct sequences and primers used for mutagenesis can be found in Supplementary Table 1, and a plasmid sequence is included as Supplementary Data 1.

### Preparation of histone octamers

Histone octamers were prepared based on literature procedures[64]. Lyophilized histones H2A, H2B, H3, and H4 were dissolved in an unfolding buffer (6 M guanidinium chloride, 20 mM Tris-HCl, 5 mM DTT, pH 7.5) at an estimated concentration of 2 mg/mL, by vortexing and incubating for 10 min on ice. The solutions were centrifuged (21,130 × g, 10 min, 4 °C), and the protein concentration of the supernatants was determined by UV-Vis spectroscopy with baseline subtraction at 320 nm. The four histones were then mixed in molar ratio 1.1:1.1:1.0:1.0, and the mixture was adjusted to 1 mg/mL total protein concentration. This mixture was dialyzed in a 3.5 kDa MW cutoff membrane (Spectra/Por 3, Spectrum) three times against 3 × 0.5 L octamer buffer (10 mM Tris-HCl, 2 M NaCl, 1 mM EDTA, 1 mM DTT, pH 7.5) for 3 h, overnight, and 3 h. Thereafter, the mixture was centrifuged (21,130 × g, 10 min, 4 °C) and the supernatant was analyzed by UV-Vis

spectroscopy and analytical HPLC (linear gradient of 0–70% B during 30 min, column #3). The supernatant was then concentrated with a 0.5 mL 30 kDa MW cut-off centrifugal filter (Amicon Ultra, Merck Millipore) and purified by SEC using octamer buffer. SEC fractions were analyzed by SDS-PAGE on 15% or 17% acrylamide gels, and fractions containing equal amounts of all four histones and eluting at the expected elution volume of octamers were mixed, centrifuged (21,130 × g, 10 min, 4 °C), concentrated using a 0.5 mL 50 kDa MW cut-off centrifugal filter (Amicon Ultra, Merck Millipore) to concentrations >20 μM, and supplemented with glycerol (ThermoFisher, cat. # 327255000) to 50% wt/wt concentration for long-term storage at −20 °C. The final samples were analyzed by analytical HPLC and SDS-PAGE to verify equal histone amounts (Supplementary Fig. 18), and by HPLC-MS to verify histone identity and molecular integrity.

### Preparation of double-stranded DNA
Double-stranded DNA with a Widom 601 nucleosome-positioning sequence[44] was prepared by polymerase chain reaction (PCR). Template (1.2 μg), primers (1.2 nmol each) and nucleotide mix (dNTP mix, 480 nmol each, New England Biolabs, cat. # N0447) were diluted on ice in ThermoPol reaction buffer (New England Biolabs, cat. # B9004S), and Taq DNA polymerase (120 units, New England Biolabs, cat. # M0273) was added for a total volume of 2.4 mL. The mixture was transferred to PCR tubes (50 μL/tube, Starlab, cat. # A1402-3700), and the PCR reaction was performed over 20 cycles with 1 min steps at 94 °C, 53 °C and 72 °C, and final 5 min incubation at 72 °C. The mixtures were then pooled, analyzed by agarose gel electrophoresis (Supplementary Fig. 18), diluted with buffer PB (QIAgen, cat. # 19066) and sodium acetate (0.1 M final concentration), and purified by QIAquick PCR purification (QIAgen protocol) using QIAprep 2.0 spin columns (QIAgen, cat. # 27115) onto buffer EB (QIAgen, cat. # 19086). Final samples containing 1.8–6.1 μM DNA concentration were stored at −20 °C.

DNA sequences, primers used for PCR, and calculated extinction coefficients can be found in Supplementary Table 2.

### Preparation of nucleosomes
Octamers were diluted 1:1 or 1:4 in high salt buffer (50 mM Tris-HCl, 2 M KCl, 1 mM DTT, pH 7.5) and kept on ice for 30 min. Then, double-stranded DNA (0.01–0.06 nmol, 1.0 equiv.) and diluted octamers (1.0–2.4 equiv.) were mixed in 2 M NaCl to a final volume of 30-80 μL and transferred to 10 kDa MW cut-off dialysis devices (Slide-A-Lyzer MINI, ThermoFisher Scientific, cat. # 69572). For larger preparative scales (1.0–1.2 nmol DNA, 1.4-1.5 mL final volume), samples were transferred to 3 mL dialysis cassettes (Slide-A-Lyzer, ThermoFisher Scientific, cat. # 66382). The devices were placed on 200 mL high salt buffer at 4 °C and dialyzed overnight into a low salt buffer (2 L, 50 mM Tris-HCl, 10 mM KCl, 1 mM DTT, pH 7.5) using a peristaltic pump (MiniPuls 3, Gilson). Then, samples were transferred to plastic microcentrifuge tubes, centrifuged (21,130 × g, 10 min, 4 °C), and analyzed by native gel electrophoresis (home-made 5% TBE gels, ran on ice at 100 V for 1.5 h with cold 0.5x TBE buffer and stained with GelRed, Supplementary Fig. 19). Titrations were performed initially to find the lowest octamer ratio at which free DNA was consumed. Then, samples prepared at such octamer ratio, and containing minimum amounts of free DNA as judged by gel electrophoresis, were combined and concentrated using 0.5 mL 50 kDa MW cut-off centrifugal filters (Amicon Ultra, Merck Millipore) to concentrations >0.8 μM (calculated based on DNA concentration). Nucleosome samples were stored at 4 °C for up to 3 months.

### Sample preparation of nucleosome-Sirt7 complexes for cryo-EM
The two complexes containing full-length SIRT7, H3K36MTU or H3K18DTU nucleosomes, and NAD+ were assembled by mixing SIRT7 and nucleosome at a 2.5:1 molar ratio (final concentrations: 10 μM SIRT7 and 4 μM nucleosomes) in the reconstitution buffer containing 25 mM HEPES-KOH, 50 mM KCl, and 300 μM NAD+, in a final volume of 300 μL. The reconstitution reaction was incubated at room temperature for 30 min and crosslinked for 10 min on ice, by adding an equal volume of the cross-linking buffer (20 mM HEPES-KOH pH 7.5, 50 mM KCl, 1 mM DTT, and 0.1% glutaraldehyde) resulting in a final glutaraldehyde concentration of 0.05%. Cross-linking was quenched by adding 60 μL of 1 M Tris pH 7.5.

For the SIRT7:H3K36MTU-nucleosome complex, 300 μL of the crosslinked complex was purified by SEC pre-equilibrated in gel filtration buffer containing 25 mM HEPES-KOH, 50 mM KCl, and 1 mM DTT. The peak fractions were analyzed by native gel electrophoresis on 6% Novex™ TBE gels (Thermo Fisher Scientific), and fractions containing SIRT7-bound nucleosomes were pooled and concentrated to an absorbance of 3 at 280 nm for cryo-EM grid preparation (Supplementary Fig. 2A). For the SIRT7:H3K18DTU-nucleosome complex, the sample was processed by gradient fixation (GraFix)[54] to stabilize the complex after cross-linking, instead of size-exclusion chromatography. The gradient contained buffer A (10% glycerol, 25 mM HEPES-KOH, and 50 mM KCl) at the top and buffer B (40% glycerol, 25 mM HEPES-KOH, 50 mM KCl, and 0.15% glutaraldehyde) at the bottom. 300 μL of the crosslinked sample was applied to the top of the gradient, followed by centrifugation (30,000 rpm, 20 h, 4 °C) on a Beckman Coulter Optima XE-90 Ultracentrifuge. Subsequently, the gradient was fractionated, and cross-linking was quenched by adding a final concentration of 100 mM Tris pH 7.5. The fractions were analyzed by native gel electrophoresis on 6% Novex™ TBE gels, and fractions containing SIRT7-bound nucleosomes were pooled, buffer-exchanged, and concentrated to an absorbance of 3 at 280 nm for cryo-EM grid preparation (Supplementary Fig. 2B).

Cryo-EM grids for the SIRT7-nucleosome complexes were prepared by applying 3 μL of concentrated sample onto 400-gold mesh R1.2/1.3 Quantifoil grids (Quantifoil Micro Tools GmbH). These grids were rendered hydrophilic by glow discharging at 15 mA for 90 s with a PELCO easiGlow device (Ted Pella Inc.). The sample was adsorbed for 30 s on the grids at 10 °C and 100% humidity, followed by blotting and plunge-freezing into liquid ethane using a Vitrobot Mark IV plunge freezer (Thermo Fisher Scientific, TFS). Cryo-EM data were collected using the automated data acquisition software EPU (TFS) on two different Titan Krios G4 transmission electron microscopes (TFS), operating at 300 kV and equipped with cold field-emission gun electron sources and either a Falcon4i direct detection camera or a Falcon4i camera with SelectrisX (see Supplementary Table 3 for further information). Datasets were recorded in counting mode at a physical pixel size of 0.726 Å and 0.83 Å at the sample level. Datasets were collected at a defocus range of 0.8 to 2.5 μm with a total electron dose of 50 e−/Å2. Image data were saved as Electron Event Recordings.

### Cryo-EM image processing, model building, and refinement
The cryo-EM image processing was performed using cryoSPARC v4.2.1[65]. The EM movie stacks were aligned and dose-weighted using patch-based motion correction (cryoSPARC implementation). Contrast transfer function (CTF) estimation was also performed using the patch-based option. For the data of the SIRT7:H3K36MTU-nucleosome complex, a blob picker was used for initial particle picking, which resulted in 792,833 particles from the initial 1000 images. These particles were used for 2D classification to generate templates for template-based particles picking on the full dataset, resulting in 7,266,736 particles. Multiple rounds of 2D classifications, ab initio reconstruction, and hetero-refinement yielded multiple 3D classes. The best 3D class comprising 1,020,323 particles was used for 3D classification with a mask covering the interaction zone between SIRT7 and the nucleosome. The best 3D classes showing distinct densities for SIRT7 were selected, resulting in 337,476 particles (Supplementary Fig. 3). Further processing of these particles by Homo refinement,

Local refinement, and non-uniform refinement resulted in a cryo-EM map at 2.8 Å in C1 symmetry (Supplementary Fig. 4 and Supplementary Table 3).

The 2D classes generated from the SIRT7:H3K36MTU-nucleosome complex were used for template picking for the SIRT7:H3K18DTU-nucleosome complex followed by two rounds of 2D classification, resulting in 1,053,851 particles which were subsequently used for ab initio reconstruction. Two 3D classes containing 703,557 particles were used for hetero-refinement, followed by 3D classification with a mask covering the interaction zone between SIRT7 and the nucleosome. Further, masked 3D classification was performed on 48,563 particles obtained from the best class from the previous round of 3D classification (Supplementary Fig. 9). This resulted in a class containing complete density for the SIRT7 domain from 22,579 particles. A final map of an overall resolution of 3.4 Å in C1 symmetry was obtained following non-uniform and local refinement (Supplementary Fig. 10 and Supplementary Table 3).

Atomic models for both structures were built by docking an AlphaFold2 (ColabFold implementation) prediction of SIRT7 and the crystal structure of the human nucleosome (PDB ID: 3LZ0)[66] into the cryo-EM map, followed by several rounds of manual rebuilding in Coot 0.9.4[67]. Real-space refinement for all built models was performed using Phenix, version 1.19.2-4158[68], using a general restraints setup (Supplementary Table 3).

### Structure visualization and analysis
Structural alignments and superpositions were performed using UCSF Chimera, UCSF ChimeraX v1.4[69] and PyMOL Version 1.8.2.0. Gel images were processed and prepared on ImageJ (version 1.53k)[70]. Figures were rendered using UCSF ChimeraX.

### Determination of SIRT7 concentration
SIRT7 aliquots were thawed, centrifuged ($21,130 \times g$, 3 min, 4 °C), and the concentration of the supernatant was estimated by Nanodrop. Then, ~0.25 μg protein were analyzed by SDS-PAGE (SurePAGE 4-12% gels, Genscript cat. # M00654, ran at 100 V and stained with Coomasie blue) together with a set of BSA standard dilutions (TFS, cat. # 23209), and bands were quantified using ImageJ. BSA data were analyzed to generate a standard curve with $r^2 > 0.98$, which was then used to calculate SIRT7 concentration. Sample preparation and SDS-PAGE were repeated twice, and the average of calculated SIRT7 concentrations was used for all further experiments.

### Electrophoretic mobility shift assays
Nucleosomes (100 nM), NAD$^+$ (0 or 150 μM), and SIRT7 (0–1.6 μM, as indicated in each experiment) were mixed in 0.5 mL low-binding microcentrifuge tubes in EMSA buffer (10 mM Tris-HCl, 50 mM KCl, 1 mM DTT, pH 7.5) for a final volume of 8 μL while kept on ice. Tubes were then transferred to a thermal shaker at 25 °C and incubated for 30 min at 800 rpm, followed by cooling on ice, addition of 50% wt/v glycerol in EMSA buffer (2 μL, 10% final glycerol concentration), and analysis by native gel electrophoresis (home-made 5% TBE gels, ran on ice at 100 V for 2 h with cold 0.25× TBE buffer and stained with

GelRed). Bands were quantified from raw TIF files using ImageJ, and data was analyzed using GraphPad Prism 10. Groups were compared by unpaired t-test for statistical analysis.

### Mass spectrometry analysis of the mechanism-based cross-link
Nucleosomes (400 or 600 nM), NAD$^+$ (150 μM), and SIRT7 (3 equiv. relative to nucleosomes) were mixed in 0.5 mL low-binding microcentrifuge tubes in EMSA buffer (10 mM Tris-HCl, 50 mM KCl, 1 mM DTT, pH 7.5) for a final volume of 20 μL while kept on ice. Tubes were then transferred to a thermal shaker at 25 °C and incubated for 30 min at 800 rpm, followed by cooling on ice and analysis by HPLC-MS on a Waters Xevo G2-XS instrument. Samples were run through a Waters Acquity BEH C4 column (1.7 μm, 100 × 2.10 mm, 300 Å) using a 15 min gradient of eluent V and eluent VI at a flow rate of 1 mL/min. Then, mass spectra corresponding to the histone H3 elution time were integrated on MassLynx and deconvoluted using MaxEnt.

### Deacetylation assays with Western blot readout
Nucleosomes (200 nM), NAD$^+$ (500 μM), and SIRT7 (0–0.8 μM) were mixed in 0.5 mL low-binding microcentrifuge tubes in deacylation buffer (10 mM Tris-HCl, 50 mM KCl, 1 mM DTT, 0.2 mg/mL BSA, pH 7.5) for a final volume of 8 μL while kept on ice. Then, tubes were transferred to a thermal shaker at 37 °C and incubated for 2 h at 800 rpm, followed by cooling on ice, mixing with 4× Laemmli sample buffer (2.7 μL, Bio-Rad, cat. #1610747, supplemented with β-mercaptoethanol), and separation by SDS-PAGE (home-made 12% or 17% gels, ran at 180 V for 0.5–1 h with Tris-glycine buffer). Gels were shortly incubated in 20% EtOH, and the samples were transferred to polyvinylidene difluoride membranes (PVDF, ThermoFisher Scientific, cat. # IB24001) using an iBlot2 western blot transfer system (ThermoFisher Scientific). After transfer, membranes were blocked in 5% skimmed milk in TBST for 1 h at room temperature, washed with TBST (2 × 1 min), and incubated with primary antibodies (Table 1) at 4 °C overnight (1:1000 or 1:2000 dilution in TBST with 2% BSA and 0.02% NaN$_3$). Then, membranes were washed with TBST (3 × 1 min), incubated with horseradish peroxidase (HRP)-conjugated secondary antibodies (Table 1) for 1 h at room temperature (1:10,000 dilution in TBST with 2% BSA), and washed with TBST (2 × 1 min) and TBS (1 × 1 min) before imaging using Clarity western enhanced chemiluminescence substrate (Bio-Rad, cat. # 1705061). Membranes were always stained first with H3 (C-term) before any Kac staining. Bands were quantified from TIF raw files using ImageJ, and data was normalized to the H3 signal and analyzed using GraphPad Prism 10. Groups were compared by unpaired t-test for statistical analysis.

### Deacylation assays with mass spectrometry readout
Nucleosomes (200 nM), NAD$^+$ (500 μM), and SIRT7 (0–0.2 μM) were mixed in 0.5 mL low-binding microcentrifuge tubes in deacylation buffer (10 mM Tris-HCl, 50 mM KCl, 1 mM DTT, 0.2 mg/mL BSA, pH 7.5) for a final volume of 8 μL while kept on ice. Then, tubes were transferred to a thermal shaker at 37 °C and incubated for 2 h at 800 rpm, followed by cooling on ice and mixing with 3.3% formic acid in H$_2$O (12 μL, 2% final concentration). Samples were mixed by vortexing,

**Table 1 | Antibodies used for deacetylation assays with western blot readout**

| Target | Organism | Vendor | Cat. # | Dilution |
|---|---|---|---|---|
| H3K18ac | Rabbit | ThermoFisher Scientific | 703896 | 1:1000 |
| H3K36ac | Rabbit | Sigma-Aldrich | SAB5600229 | 1:1000 |
| H3 (C-term) | Mouse | Cell Signaling Technology | 3638 | 1:2000 |
| SIRT7 (N-term) | Rabbit | Cell Signaling Technology | 5360 | 1:1000 |
| Mouse IgG (HRP-conjugated) | Horse | Cell Signaling Technology | 7076 | 1:10,000 |
| Rabbit IgG (HRP-conjugated) | Goat | Cell Signaling Technology | 7074 | 1:10,000 |

centrifuged (21,130 × *g*, 3 min, 4 °C), and analyzed by HPLC-MS on a Waters Xevo G2-XS instrument. Samples were run through a Waters Acquity BEH C4 column (1.7 μm, 100 × 2.10 mm, 300 Å) using 15 min gradients of eluent V and eluent VI at a flow rate of 1 mL/min. Then, chromatograms of the indicated ions (Table 2) were extracted and exported to GraphPad Prism 10, where they were smoothed, integrated, and analyzed. Baseline correction for statistical analysis of SIRT7 deacylation was performed against reactions without SIRT7.

## Preparation of plasmids for mammalian cell expression

A pcDNA4/TO plasmid with cloned human *SIRT7* and a C-terminal HA tag was received from the Vaquero laboratory at the Josep Carreras Leukemia Research Institute. The initial HA tag was excised through digestion with NotI (New England Biolabs, cat. # R3189S) and XBaI HF (New England Biolabs, cat. # R0145S) and subsequent gel purification. Then, it was replaced by a linker-mCherry(Uniprot X5DSL3)-HA tag construct (Twist Bioscience) via Gibson assembly (New England Biolabs, cat. # E2611). The final plasmid was isolated through rounds of transformation into *E. coli* XL10-Gold (Agilent, cat. # 200315), colony PCR, colony overnight culture, and 'miniprep' DNA extraction (QIAGEN, cat. # 27106×4). The plasmid sequence encoding SIRT7(wild type)-mCherry-HA can be found as a supplementary file. Thereafter, mutations were introduced by site-directed mutagenesis with the Q5 polymerase (New England Biolabs, cat. # M0491) and primers as indicated below, followed by KLD enzyme mix treatment (New England Biolabs, cat. # M0554). All plasmids were transformed into *E. coli* XL10-Gold and purified by 'midiprep' DNA extraction (QIAGEN, cat. # 12143).

All construct sequences and primers used for mutagenesis can be found in Supplementary Table 4, and a plasmid sequence is included as Supplementary Data 1.

## Cell culture and Western blot

Human embryonic kidney 293 F (HEK293F) *SIRT7*[−/−] cells were obtained from the Vaquero laboratory (Josep Carreras Leukemia Research Institute) and maintained as adherent monolayers in Dulbecco's modified Eagle medium (DMEM, ThermoFisher Scientific, cat. # 41966) with 10% fetal bovine serum (FBS, Sigma-Aldrich, cat. # F9665), MEM non-essential amino acids (ThermoFisher Scientific, cat. # 11140050) and penicillin/streptomycin (ThermoFisher Scientific, cat. # 15140), at 37 °C with 5% $CO_2$ in a humidified atmosphere. Cells were typically maintained in T75 flasks (TPP Techno Plastic Products AG, cat. # 90075) up to ~80% confluency, and 0.4–0.8 million cells passaged to a new T75 flask, up to passage P11-P15. For passaging, medium was removed, cells were washed with PBS (ThermoFisher Scientific, cat. # 10010) and detached with trypsin-EDTA (0.05%, phenol red, ThermoFisher Scientific, cat. # 25300). Then, the cell dispersion was diluted in DMEM, centrifuged (500 × *g*, 3 min, 25 °C), and the pellet was resuspended in DMEM for manual cell counting and transfer to the new flask.

For test experiments and Western blot, 50,000 cells/well were seeded into 24-well plates (TPP Techno Plastic Products AG, cat. # 92012) in 800 μL DMEM, incubated for 24 h or until 70% confluency, and treated with a mixture of plasmid and lipofectamine 2000 (ThermoFisher Scientific, cat. # 11668) in DMEM without antibiotics (25 μL/well pre-incubated for 5 min at room temperature, containing 800 ng plasmid and 3:1 lipofectamine ratio). After incubation overnight, cells were either imaged or the media were removed and the cells were washed with PBS (2 × 300 μL/well) and treated with HEK293F lysis buffer (100 μL-well, 50 mM Tris-HCl pH 7.4, 150 mM NaCl, 1% Triton X-100, 0.5% sodium deoxycholate, 0.1% SDS, with cOmplete protease inhibitors) at 4 °C for 15 min. The lysates were collected, sonicated in a water bath (3 × 30 s on/30 s off) and centrifuged (21,130 × *g*, 10 min, 4 °C), and the supernatants were mixed with Laemmli buffer and stored at −20 °C until use. Sample protein concentrations were estimated using the bicinchoninic assay (BCA assay,

Merck, cat. # 71285-3), and 20 μg of the samples were analyzed by SDS-PAGE (home-made 12% gels, ran at 120 V for 15 min and then 170 V for 1 h with Tris-glycine buffer). Samples were then transferred to PVDF membranes, stained (Table 3), and imaged as before.

## Immunofluorescence and data analysis

8-well μ-slides with high poly-L-lysine (ibidi, cat. # 80804) were further coated with poly-L-lysine (0.1% solution, 40 μL/well, Electron Microscopy Sciences, cat. # 19320-B) at room temperature for 1 h, washed with PBS (2 × 200 μL/well) and left to dry for 2 h. Then, 25000 cells were transferred to each well in 300 μL DMEM, incubated for 24 h, and treated with a mixture of plasmid and lipofectamine 2000 in DMEM without antibiotics (25 μL/well pre-incubated for 5 min at room temperature, containing 400 ng plasmid and 3:1 lipofectamine ratio). After incubation overnight (~20 h), media were removed, and the cells were washed with PBS (3 × 150 μL/well) and treated with paraformaldehyde (Electron Microscopy Sciences, cat. # 15714) diluted to 4% in PBS for 10 min at room temperature. Solutions were removed, the cells were washed with PBS (1 × 150 μL/well), treated with 1% Triton X-100 (Sigma-Aldrich, cat. # X-100) in PBS for 15 min at 4 °C, and washed with PBS (1 × 150 μL/well). Cells were then blocked in PBS with 5% BSA (Merck, cat. # A7906) and 0.1% Triton X-100 for 2 h at room temperature, washed with PBS (1 × 150 μL/well), and incubated with primary antibodies (Table 4) for 2 h at room temperature (1:200 dilution in PBS with 2% BSA and Triton X-100 as indicated below). Thereafter, cells were washed with PBS (3 × 3 min, 150 μL/well, with Triton X-100 as indicated below), incubated with secondary antibody (Table 4) and Hoechst 33342 (Chemodex, cat. # B0030) for 1 h at room temperature (1:1000 dilution, or 1:300 for H3K36ac, in PBS with 1 μg/mL Hoechst, 2% BSA and Triton X-100 as indicated below). Finally, cells were washed with PBS (1 × 3 min, 150 μL/well, with Triton X-100 as indicated below), and with PBS only (2 × 3 min, 150 μL/well) and stored at 4 °C for up to 24 h before imaging. Fluorescence images were recorded on a CSU-W1 spinning disk confocal microscope (Nikon Instruments Inc.) on an Eclipse Ti2-E motorized stand and with a CFI Plan Apo Lambda 60x oil-immersion objective and a Prime 95B camera (Teledyne Photometrics).

**Table 2 | Mass of extracted ions for each nucleosome and analysis type**

| Nucleosome | Type of analysis | Ion identity | m/z |
|---|---|---|---|
| H3K18ac | Substrate | $[M + 21H]^{21+}$ | 729.52 |
| H3K18ac | Product | $[M + 21H]^{21+}$ | 727.52 |
| H3K18dec | Substrate | $[M + 21H]^{21+}$ | 734.86 |
| H3K18dec | Product | $[M + 21H]^{21+}$ | 727.52 |
| H3K36ac | Substrate | $[M + 21H]^{21+}$ | 731.05 |
| H3K36ac | Product | $[M + 21H]^{21+}$ | 729.04 |
| H3K36dec | Substrate | $[M + 21H]^{21+}$ | 736.39 |
| H3K36dec | Product | $[M + 21H]^{21+}$ | 729.04 |

**Table 3 | Antibodies used for cell lysate western blots**

| Target | Organism | Vendor | Cat. # | Dilution |
|---|---|---|---|---|
| SIRT7 (N-term) | Rabbit | Cell Signaling Technology | 5360 | 1:1000 |
| mCherry | Rabbit | Proteintech | 26765-1-AP | 1:1000 |
| GAPDH | Rabbit | Cell Signaling Technology | 2118 | 1:1000 |
| Mouse IgG (HRP-conjugated) | Horse | Cell Signaling Technology | 7076 | 1:10,000 |
| Rabbit IgG (HRP-conjugated) | Goat | Cell Signaling Technology | 7074 | 1:10,000 |

**Table 4 | Antibodies used for immunofluorescence assays**

| Target | Organism | Vendor | Cat. # | Dilution | [Triton X-100] |
|---|---|---|---|---|---|
| H3K18ac | Rabbit | ThermoFisher Scientific | MA5-24669 | 1:200 | 0.1% |
| H3K36ac | Rabbit | ThermoFisher Scientific | MA5-24672 | 1:200 | - |
| H4K16ac | Rabbit | ThermoFisher Scientific | MA5-34717 | 1:200 | 0.1% |
| Rabbit IgG (Alexa Fluor 488-conjugated) | Donkey | ThermoFisher Scientific | A-21206 | 1:1000 or 1:300 | as above |

Images were batch-processed using the ImageJ Macro in Fiji v2.14.0, where background fluorescence was subtracted, cell nuclei were masked according to Hoechst staining (automatic threshold Huang dark), and the mean intensity from each channel as well as nuclear area was calculated. Data were then analyzed with RStudio vR-4.4.1, where cell nuclei with areas outside of the 20–270 $\mu m^2$ range were excluded from analysis, and transfected nuclei were selected based on mCherry signal within 25–780 mean intensity (after background subtraction). For each image, the Alexa Fluor 488 mean intensity was corrected for different staining efficiency across experiments:

$$\text{Corrected mean intensity} = \frac{\text{Mean intensity (individual nucleus)}}{\text{Median mean intensity (mCherry} - \text{negative nuclei)}}$$

The corrected mean intensities were represented with GraphPad Prism 10 as violin plots truncated to the experimental values, and analyzed through the non-parametric, two-tailed, Kruskal–Wallis and Dunn's multiple comparisons tests.

### Reporting summary

Further information on research design is available in the Nature Portfolio Reporting Summary linked to this article.

## Data availability

The reconstructed cryo-EM maps and atomic models generated in this study have been deposited in the EMDB and PDB databases, under accession codes EMD-51453 and PDB 9GMR for SIRT7:H3K36MTU, and under accession codes EMD-51449 and PDB 9GMK for SIRT7:H3K18DTU. The micrograph image data have been deposited in the EMPIAR database under accession code EMPIAR-12525. All other data generated and analyzed in this study are provided in the Supplementary Information/Source Data file. Plasmid sequences are provided as Supplementary Data 1. Source data are provided with this paper.

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

## Acknowledgements

We would like to thank Dr. Julie Bolding and Prof. Christian A. Olsen (University of Copenhagen) for sharing of reagents, Dr. Kelvin Lau (EPFL) for assistance with the optimization of SIRT7 production, Prof. Alejandro Vaquero and Dr. Ana Marazuela (Josep Carreras Leukemia Research Institute) for sharing of *SIRT7*⁻/⁻ cell lines and mammalian expression plasmids, and Prof. Thomas Schalch for comments on the

manuscript. The cryo-EM data collection was performed at the Dubochet Center for Imaging Lausanne (a common initiative from EPFL, UNIGE, UNIL, UNIBE) with the help of A. Myasnikov, B. Beckert, S. Nazarov, I. Mohammed and E. Uchikawa. This work was supported by the Independent Research Fund Denmark (DFF International Postdoctoral Grant #2028.00011B to C.M.-Y.), the Swiss National Science Foundation (SNSF Swiss Postdoctoral Fellowship #TMPFP2_217187 to C.M.-Y., grant 310030_200604 to B.F., and grant IZLCZO_206089 to D.N. and H.S.), the Swiss Chemical Society Foundation (Alfred Werner scholarship to P.N.F.), "la Caixa" Foundation (ID 100010434, fellowship LCF/BQ/EU22/11930058 to E.C.-S.), and EPFL.

## Author contributions

Conceptualization, C.M.-Y. and B.F.; methodology, C.M.-Y., B.E.E., P.N.F., and E.C.-S.; investigation, C.M.-Y., B.E.E., P.N.F., and D.N.; resources, C.M.-Y., P.N.F., and E.C.-S.; formal analysis, C.M.-Y., B.E.E., P.N.F., D.N., and B.F.; writing–original draft, C.M.-Y.; writing–review & editing, all authors; visualization, C.M.-Y. and B.E.E.; supervision, C.M.-Y., H.S., and B.F.; funding acquisition, C.M.-Y., D.N., H.S. and B.F.

## Competing interests

The authors declare no competing interests.
