## [Transparent Peer Review file · Nature Communications]

Structural basis of SIRT7 nucleosome engagement and substrate specificity

Corresponding Author: Professor Beat Fierz

Version 0:

Reviewer comments:

Reviewer #1

(Remarks to the Author)

In the manuscript entitled “Structural basis of SIRT7 nucleosome engagement and substrate specificity”, Beat et al. has explored the structural specifics of Sirt7’s ability to deacetylate H3 lysine 36 (H3K36) or H3K18 within nucleosomes. The methodologies used include cryo-EM, mutagenesis, and enzymatic assays. They determined the cryo-EM structures of SIRT7 interacting with the nucleosome, focusing on two major substrate residues, H3K36 and H3K18. They found out that SIRT7 doesn’t depend on a singular anchor but sets up a multivalent interaction surface throughout the nucleosome via a unique nucleosome-binding domain. The research also identified the unique characteristics of the N-terminus of SIRT7, which extends the nucleosome-binding domain and spans across the nucleosomal surface to the acidic patch. The catalytic domain of SIRT7 is located at the H3-tail exit site, where it engages both DNA gyres of the nucleosome. Additionally, the binding pose of the enzyme shifts when contacting H3K36 versus H3K18, leading to structural changes in both SIRT7 and the nucleosome. In conclusion, this study adds to the body of knowledge in nucleosome modification studies by offering a new structural insight into how chromatin-modifying enzymes like SIRT7 achieve substrate specificity. I support publication in Nature Communication after authors address the following issues:

1. The difference between the two complex structures (Fig. 3C in page 8 and the main text in page 7) was claimed by the authors to be functional for H3K18 targeting. However, the results shown here could not rule out that the structure differences for both the catalytic domain of SIRT7 and the terminal DNA are from the intrinsic dynamics of the molecules themselves just as the dynamics of SIRT6 revealed recently in an eLife paper (Smirnova, E. et al. 12:RP87989 (2024).). The key is whether K36Ac and K18Ac share a similar conformation to work but different conformation distributions on nucleosome (different conformation may have different activities)?
2. The main text of the 4th line in page 6, and the supplementary figure 6 in page S8 in the SI file are not consistent. The authors should place some structures for H3K36me3 reader domains for comparison or cite the supplementary figure 6 elsewhere.
3. Page 7 line 1 in the main text, the authors claimed that the loop 120-138 is predicted at lower confidence, however, no data was shown for this kind of confidence. The authors should present the pLDDT and PAE in the supplementary figure 7 as evidence.
4. Page 7 line 7 (and Fig. 2B in page 5) in the main text, a part of C-terminal tail in the structure (the one for H3K18DTU) is visible, please specify the residue number for clarity.
5. Page 7 line 18 in the main text, how does one measure the average resolution by a local resolution map? Use terms like “a medium resolution” would be more appropriate.
6. Page 9 line 17 and line 20 in the main text, the figure number is incorrect. Fig. 3C should be Fig. 3E, and Fig. 3D should be Fig. 3F.
7. Page 21 line 7 in the main text, and the supplementary table 1 in page S13 in the SI file, the configuration for Falcon 4 and Falcon 4i with SelectrisX is dramatically different on performance, please clarify the camera name as the potential reference in later works.
8. Page 21 line 12 in the main text and the supplementary figure S3 in page S5 and supplementary figure S8 in page S9 in the SI file, please unify the cryoSPARC version.

Reviewer #2

(Remarks to the Author)

In this submission, Moreno-Yruela et al., determined two structures of nucleosome-bound histone deacetylase SIRT7, one with H3K18DTU and one with H3K36MTU. SIRT7 is a member of an extended Sirtuin family composed of several members in Eukaryotes. This family of proteins play a critical role in DNA damage repair, inflammation and aging, and have been long pursued as a small molecule target for cancer and degenerative diseases. Different members of this family target different histone modifications, and thus, it is essential to understand the mechanisms of their selectivity. One of the key approaches is to probe the system using structural biology approaches coupled with activity assays, mutagenesis studies, etc.

This is a very well-rounded publication, with essentially any major comment that might accompany a publication of this kind having already been addressed by the authors in their experimentation and the write-up. This is a solid work that sheds light on nucleosome binding and interactions leading up to deacylation with both modifications. It is very interesting to see how SIRT7 compares to another Sirtuin, SIRT6, revealing a surprisingly completely different mode of interaction with the nucleosome.

There are small comments that this reviewer has:

1. Can the authors comment whether the use of crosslinking (crosslinked H3K18DTU vs non-crosslinked H3K36MTU samples) could have an effect on the differences seen?
2. Can the authors comment on whether the preferred orientation in the SIRT7:H3K18DTU subset could have an effect on their interpretation of the mechanism of histone lysine selectivity?
3. In Figure 2g (left), could the authors remake the SIRT6-nucleosome complex picture to reflect the same orientation and view as that of SIRT7 from Figure 2b (left)? This would increase the impact of this picture in the figure, as it would provide a more precise and direct correlation between the two Sirtuins?

Reviewer #3

(Remarks to the Author)

The manuscript by Fierz et al. presented the structural characterization of SIRT7 bound with two nucleosomes, one with modification at K36 and the other K18. Associated biochemical characterizations of different mutant SIRT7 enzymes were done as well to assist the understanding of the structural observations. The study was from a very productive young investigator who has produced quite a number of chromatin-related publications. All experiments reported in the manuscript had high quality presented data. So there is no concern about the rigor of the study. The two cryo-EM structures are important additions to the existing chromatin structure pool. They also support published results about SIRT7 activity and its dependence on the chromatin context for activity. I would like to support its publication after the authors address some concerns discussed below.

1. The authors will need to provide some chemistry evidence to show that the proposed bicyclic intermediate shown in Figure 2C does exist and it doesn't repel the methylamine. From chemistry perspective, this intermediate that has one carbon connected to four hetero and electronegative atoms is not going to be stable. The JACS paper by Cole et al. reported a thioimidate final product. That structure makes more sense from chemistry perspective. The cryo-EM structure cannot provide definite assignment of atoms. The map showing in Figure 2C is not convincing to definitely say that is a methylamine there. A biochemistry assay to show there is no methylamine formed is a key test. It needs to be provided.
2. The authors implied at multiple occasions that the less defined structure for SIRT7 bound to K18ac-nucleosome supports the enzyme's low activity toward this site. Instead of implying it, the authors need to use a more definitive tone. That is what makes this paper interesting.
3. The mutagenesis results are confusing. Instead of aiding the structural results, some are actually against the cryo-EM structural observation. The results for SIRT7(R11A/K12A) are puzzling. If their interactions with the acidic patch contributes to the overall binding of SIRT7 to chromatin, decreased activity should be observed. Otherwise, it could just indicate the observed cryo-EM structure is just an artificial observation. The authors need to provide a more convincing argument.
4. The data for SIRT7(K272A/K275A/K276A) are even more confusing. Yes, the argument provided by the authors can explain why this mutant is more activity to deacetylate K18. But this mutant as shown in Figure 5B is also more active toward K36. This doesn't make any sense. A better explanation is needed.

All concerns provide above are based on trust of high rigor of all experiments. The authors may check some of the results to see if some of these confusing observations were due to experimental errors. Nevertheless, this is an important piece of study.

Version 1:

Reviewer comments:

Reviewer #1

(Remarks to the Author)

All of my concerns have been addressed. I support the publication of this work.

Reviewer #2

(Remarks to the Author)

The resubmitted manuscript addressed all the issues and request that this reviewer (#2) had in the prior review. The work that the authors did to answer other reviewer' concerns adds to the value of the manuscript. I have no further concerns

Reviewer #3

(Remarks to the Author)

The authors have addressed most concerns from this reviewer. The new manuscript is ready for acceptance.

REVIEWER COMMENTS

Reviewer #1 (Remarks to the Author):

In the manuscript entitled “Structural basis of SIRT7 nucleosome engagement and substrate specificity”, Beat et al. has explored the structural specifics of Sirt7's ability to deacetylate H3 lysine 36 (H3K36) or H3K18 within nucleosomes. The methodologies used include cryo-EM, mutagenesis, and enzymatic assays. They determined the cryo-EM structures of SIRT7 interacting with the nucleosome, focusing on two major substrate residues, H3K36 and H3K18. They found out that SIRT7 doesn't depend on a singular anchor but sets up a multivalent interaction surface throughout the nucleosome via a unique nucleosome-binding domain. The research also identified the unique characteristics of the N-terminus of SIRT7, which extends the nucleosome-binding domain and spans across the nucleosomal surface to the acidic patch. The catalytic domain of SIRT7 is located at the H3-tail exit site, where it engages both DNA gyres of the nucleosome. Additionally, the binding pose of the enzyme shifts when contacting H3K36 versus H3K18, leading to structural changes in both SIRT7 and the nucleosome. In conclusion, this study adds to the body of knowledge in nucleosome modification studies by offering a new structural insight into how chromatin-modifying enzymes like SIRT7 achieve substrate specificity. I support publication in Nature Communication after authors address the following issues:

1. The difference between the two complex structures (Fig. 3C in page 8 and the main text in page 7) was claimed by the authors to be functional for H3K18 targeting. However, the results shown here could not rule out that the structure differences for both the catalytic domain of SIRT7 and the terminal DNA are from the intrinsic dynamics of the molecules themselves just as the dynamics of SIRT6 revealed recently in an eLife paper (Smirnova, E. et al. 12:RP87989 (2024).). The key is whether K36Ac and K18Ac share a similar conformation to work but different conformation distributions on nucleosome (different conformation may have different activities)?

We would like to thank the reviewer for pointing out this concept, as we believe it is key for understanding the activity of SIRT6 and SIRT7 on the nucleosome. Similar to SIRT6, as shown in the cited article, our data indicates that SIRT7 has multiple nucleosome-bound conformations. Moreover, different conformations target distinct residues on the H3 tail, as revealed by our mechanism-based capture method, which isolates a substrate-specific conformational subset. To highlight this concept and relate it to the SIRT6 literature, we have extended the discussion with:

“Both SIRT6 and SIRT7 target the H3 tail while remaining in contact with the nucleosome acidic patch. Their nucleosome-bound states remain dynamic – the enzymes sample a conformational landscape, allowing them to reach distinct substrate sites thereby conferring each enzyme with a distinct substrate selectivity (ref: Smirnova, 2024). Here, we capture SIRT7 in both H3K18- and H3K36-targeting conformations, corresponding to two functional states within this conformational landscape.”

2. The main text of the 4th line in page 6, and the supplementary figure 6 in page S8 in the SI file are not consistent. The authors should place some structures for H3K36me3 reader domains for comparison or cite the supplementary figure 6 elsewhere.

We agree with the reviewer that Supplementary Fig. 6 (now Supplementary Fig. 7) was not fully informative. We have now added a comparison of nucleosomal H3K36 binding by a PWWP domain and by SIRT7:

**Supplementary Fig. 7. A**, Surface potential of SIRT7 with highlighted nucleosome-binding regions. **B**, Comparison of nucleosome binding at H3K36 by a PWWP domain (PDB 6S01) and SIRT7.

3. Page 7 line 1 in the main text, the authors claimed that the loop 120-138 is predicted at lower confidence, however, no data was shown for this kind of confidence. The authors should present the pLDDT and PAE in the supplementary figure 7 as evidence.

We agree that this information is useful to interpret the AF3 data properly. We have updated Supplementary Fig. 7 (now Supplementary Fig. 8) with plots of the pLDDT confidence of the loop and its environment, and of the overall PAE matrix of the predicted complex:

Supplementary Fig. 8. AlphaFold 3 prediction of the SIRT7:nucleosome complex. **A**, Overlay of the AF3 SIRT7:nucleosome complex and our experimental model of SIRT7 bound to the H3K36MTU nucleosome (not shown). Structures were aligned by the histone octamer. **B**, Detail of the flexible loop within the catalytic domain (aa 120–138) and its AF3-predicted interaction with DNA. This region has lower confidence than the rest of the catalytic domain according to AF3, as seen in the lower panel colored by pLDDT (created with PAE viewer). **C**, Predicted aligned error (PAE) matrix of the AF3 complex, created with PAE viewer.

4. Page 7 line 7 (and Fig. 2B in page 5) in the main text, a part of C-terminal tail in the structure (the one for H3K18DTU) is visible, please specify the residue number for clarity.

We thank the reviewer for pointing out the missing information. We have added the residue numbers to the text:

“In our model, a part of it (aa 331–362) is visible contacting the top of the catalytic domain (Fig. 2B).”

5. Page 7 line 18 in the main text, how does one measure the average resolution by a local resolution map? Use terms like “a medium resolution” would be more appropriate.

We agree with the change suggested by the reviewer and have modified the text accordingly:

“Nonetheless, we obtained a cryo-EM map of the SIRT7: H3K18DTU complex at a medium resolution (3–7.5 Å, and 3.5 Å overall, Fig. 3A and Supplementary Figs. 9–10)”

6. Page 9 line 17 and line 20 in the main text, the figure number is incorrect. Fig. 3C should be Fig. 3E, and Fig. 3D should be Fig. 3F.

We thank the reviewer for noticing these mistakes. They have been corrected.

7. Page 21 line 7 in the main text, and the supplementary table 1 in page S13 in the SI file, the configuration for Falcon 4 and Falcon 4i with SelectrisX is dramatically different on performance, please clarify the camera name as the potential reference in later works.

We thank the reviewer for noticing this mistake. We have updated the methods in the main text to indicate that two Titan Krios G4 were used, one with a Falcon4i camera and the other with a Falcon4i camera with SelectrisX.

8. Page 21 line 12 in the main text and the supplementary figure S3 in page S5 and supplementary figure S8 in page S9 in the SI file, please unify the cryoSPARC version.

We have also updated the methods in the main text to fix this inconsistency and indicate that cryoSPARC v4.2.1 was used.

Reviewer #2 (Remarks to the Author):

In this submission, Moreno-Yruela et al., determined two structures of nucleosome-bound histone deacetylase SIRT7, one with H3K18DTU and one with H3K36MTU. SIRT7 is a member of an extended Sirtuin family composed of several members in Eukaryotes. This family of proteins play a critical role in DNA damage repair, inflammation and aging, and have been long pursued as a small molecule target

for cancer and degenerative diseases. Different members of this family target different histone modifications, and thus, it is essential to understand the mechanisms of their selectivity. One of the key approaches is to probe the system using structural biology approaches coupled with activity assays, mutagenesis studies, etc.

This is a very well-rounded publication, with essentially any major comment that might accompany a publication of this kind having already been addressed by the authors in their experimentation and the write-up. This is a solid work that sheds light on nucleosome binding and interactions leading up to deacylation with both modifications. It is very interesting to see how SIRT7 compares to another Sirtuin, SIRT6, revealing a surprisingly completely different mode of interaction with the nucleosome.

There are small comments that this reviewer has:

1. Can the authors comment whether the use of crosslinking (crosslinked H3K18DTU vs non-crosslinked H3K36MTU samples) could have an effect on the differences seen?

We thank the reviewer for pointing out this detail. We have extended the discussion to address it:

“This second conformation proved to be less defined than the previous H3K36-bound state and required additional crosslinking by gradient fixation for cryo-EM, which might indicate that the putative H3K18-bound conformational landscape is even broader.”

2. Can the authors comment on whether the preferred orientation in the SIRT7:H3K18DTU subset could have an effect on their interpretation of the mechanism of histone lysine selectivity?

We thank both reviewer 2 and reviewer 3 for their comments on the mechanism of selectivity of SIRT7. We have changed the language across the manuscript to make it more definitive that the two orientations we observe reveal these mechanisms, including in the results:

“In summary, our results indicate that nucleosome binding directs SIRT7 activity preferably towards deacetylation of H3K36, and that access to the H3K18 substrate is disfavoured, as it requires the adoption of a more dynamic conformation.”

in the discussion:

“In this configuration, SIRT7:DNA contacts are relaxed and the exiting linker DNA is bent, possibly due to increased space requirement of the H3 tail and interactions with the reoriented catalytic domain. These conformational differences to the canonical nucleosome structure and the increase in structural heterogeneity of the complex provide a mechanistic explanation for the lower activity of SIRT7 towards H3K18 deacylation.”

and in the concluding paragraph:

“Moreover, the structures reveal that distinct conformations of the enzyme active site on the nucleosome, as well as their respective dynamics, drive substrate specificity within the histone tails, as opposed to the recognition of the peptide sequence context.”

3. In Figure 2g (left), could the authors remake the SIRT6-nucleosome complex picture to reflect the same orientation and view as that of SIRT7 from Figure 2b (left)? This would increase the impact of this picture in the figure, as it would provide a more precise and direct correlation between the two Sirtuins?

We thank the reviewer for this suggestion. We have changed the panel to show the same orientation of the nucleosome as in Figure 2B.

Reviewer #3 (Remarks to the Author):

The manuscript by Fierz et al. presented the structural characterization of SIRT7 bound with two nucleosomes, one with modification at K36 and the other K18. Associated biochemical characterizations of different mutant SIRT7 enzymes were done as well to assist the understanding of the structural observations. The study was from a very productive young investigator who has produced quite a number of chromatin-related publications. All experiments reported in the manuscript had high quality presented data. So there is no concern about the rigor of the study. The two cryo-EM structures are important additions to the existing chromatin structure pool. They also support published results about SIRT7 activity and its dependence on the chromatin context for activity. I would like to support its publication after the authors address some concerns discussed below.

1. The authors will need to provide some chemistry evidence to show that the proposed bicyclic intermediate shown in Figure 2C does exist and it doesn't repel the methylamine. From chemistry perspective, this intermediate that has one carbon connected to four hetero and electronegative atoms is not going to be stable. The JACS paper by Cole et al. reported a thioimidate final product. That structure makes more sense from chemistry perspective. The cryo-EM structure cannot provide definite assignment of atoms. The map showing in Figure 2C is not convincing to definitely say that is a methylamine there. A biochemistry assay to show there is no methylamine formed is a key test. It needs to be provided.

We thank the reviewer for suggesting these experiments, as they provided a more definitive model of the mechanism-based crosslink. To address the identity of this crosslink within SIRT7, we first modelled either the bicyclic catalytic intermediate or the thioimidate shown by Cole et al. in a recent SIRT6 structure (here referred to as bicyclic imine link) into our cryo-EM density. This revealed that a portion of the density could only be explained by a methylamine substituent being present. Then, we analysed the mechanism-based complexes of H3K36MTU and H3K36DTU nucleosomes with SIRT7 by LC-MS and found only the mass that corresponds to the catalytic intermediates and not to the alternative thioimidate species. These experiments served to confirm that SIRT7 does not lead to elimination of the amine substituent. We updated the main text to reflect this and added Supplementary Figure 6:

“The map permitted reconstruction of the mechanism-based adenosine diphosphate ribose (ADPr)-MTU conjugate (Fig. 2C), and the observed density and LC-MS analysis were found to be consistent with the bicyclic intermediate also observed in the X-ray structure of SIRT5 bound to a thioamide (see also Supplementary Fig. 6).”

Supplementary Fig. 6. A, Fitting of a bicyclic catalytic intermediate or a bicyclic imine link model into the H3K36MTU adduct density, and specific density corresponding to the methylamine substituent. Here, our cryo-EM density was less compatible with the imine geometry and clearly indicated that the methylamine substituent is still present. **B**, LC-MS analysis of H3K36MTU and H3K36DTU incubated without (top spectra) and with (bottom spectra) SIRT7, and highlighted mass of the ADPr adducts, which include the methylamine and decylamine substituents, respectively. This analysis shows the methylamine and decylamine substituents likely as 1'-isothiuronium adducts, but does not indicate any imine species. Together, we confirmed that thioureas form the envisioned mechanism-based complexes within the SIRT7 active site via the equilibrium of 1'-isothiuronium and bicyclic catalytic intermediates without amine elimination. Under our LC-MS conditions, H3 co-eluted partially with H2A.

2. The authors implied at multiple occasions that the less defined structure for SIRT7 bound to K18ac-nucleosome supports the enzyme's low activity toward this site. Instead of implying it, the authors need to use a more definitive tone. That is what makes this paper interesting.

As mentioned above, we thank both reviewer 2 and reviewer 3 for their comments on the mechanism of selectivity of SIRT7. We have changed the language across the manuscript to make it more definitive that the two orientations we observe reveal these mechanisms, including in the results:

“In summary, our results indicate that nucleosome binding directs SIRT7 activity preferably towards deacetylation of H3K36, and that access to the H3K18 substrate is disfavoured, as it requires the adoption of a more dynamic conformation.”

in the discussion:

“In this configuration, SIRT7:DNA contacts are relaxed and the exiting linker DNA is bent, possibly due to increased space requirement of the H3 tail and interactions with the reoriented catalytic domain. These conformational differences to the canonical nucleosome structure and the increase in structural heterogeneity of the complex provide a mechanistic explanation for the lower activity of SIRT7 towards H3K18 deacylation.”

and in the concluding paragraph:

“Moreover, the structures reveal that distinct conformations of the enzyme active site on the nucleosome, as well as their respective dynamics, drive substrate specificity within the histone tails, as opposed to the recognition of the peptide sequence context.”

3. The mutagenesis results are confusing. Instead of aiding the structural results, some are actually against the cryo-EM structural observation. The results for SIRT7(R11A/K12A) are puzzling. If their interactions with the acidic patch contributes to the overall binding of SIRT7 to chromatin, decreased activity should be observed. Otherwise, it could just indicate the observed cryo-EM structure is just an artificial observation. The authors need to provide a more convincing argument.

We thank the reviewer for pointing this out. Our data show that the interaction of SIRT7 with the arginine-anchor binding site, through R11 and K12, is not driving overall binding and activity on nucleosomes, as SIRT7(R11A, K12A) does not show lower binding (Fig. 4B) or lower activity (Fig. 4D) on nucleosomes than wild type. To make this observation clearer, we have added statistical analysis (ns) to the comparison of wild type and this mutant on EMSA experiments:

Instead, we identified through the cryo-EM structures that the entire $\alpha 1$ helix is necessary for multivalent interaction with the histone octamer and specific recognition of nucleosomes, as demonstrated by the lower binding affinity of the truncated mutant SIRT7(40-400) (Fig. 4B). This mutant is recruited solely by non-specific interactions with DNA and presents lower deacetylase

activity when challenged with free DNA (Fig. 4E). To further demonstrate this effect under native conditions, we transfected HEK293F *SIRT7*^{-/-} cells with SIRT7(40-400) or wild type and observed that the truncated mutant is less active than wild type within the cell nucleus, in particular against H3K18ac targets. We have added Fig. 4F (and Supplementary Fig. 13, see below) showing this effect:

Fig. 4. Role of the N-terminal domain to nucleosome binding and deacetylation. (...) f Violin plot (truncated) of H3K18ac and H3K36ac levels in HEK293F *SIRT7*^{-/-} cells upon transient transfection with SIRT7-mCherry constructs as indicated, measured by immunofluorescence. Internal lines represent median and quartile values of $n \geq 4$ distinct experiments (ns: $p > 0.05$) and light background line indicates median value for WT SIRT7. See Supplementary Fig 13 for SIRT7 Western blots, representative microscopy images and control H4K16ac quantification.

and extended the results section with the following:

“Based on these results, we hypothesised that the role of the SIRT7 N-terminal domain lies in its specific recruitment to nucleosomes over other nucleic acids. We thus challenged SIRT7 with increasing amounts of competitor DNA, while monitoring its deacetylation activity on nucleosomes. Indeed, SIRT7(40–400) was efficiently outcompeted by free DNA, while the wild-type enzyme retained its activity independent of added competitor DNA (Fig. 4E). Next, we transfected a stable HEK293F *SIRT7*^{-/-} cell line with SIRT7 wild type or SIRT7(40–400) and found that the wild type induced lower acetylation levels than the truncated construct, especially against H3K18ac marks (Fig. 4F and Supplementary Fig. 13). This was consistent with SIRT7(40–400) being a less efficient deacetylase within the cellular environment. Thus, the full nucleosome-binding domain is necessary for efficient chromatin recruitment of SIRT7 in a complex environment.”

4. The data for SIRT7(K272A/K275A/K276A) are even more confusing. Yes, the argument provided by the authors can explain why this mutant is more activity to deacetylate K18. But this mutant as shown in Figure 5B is also more active toward K36. This doesn't make any sense. A better explanation is needed.

We agree with the reviewer that the data we present sometimes requires a complex explanation and might appear confusing at times. To address this and improve the robustness of our conclusions and analyses, we have (1) updated Figure 5B to show a H3K36 western blot that is more representative of the overall data shown in Figure 5C:

and (2) we transfected HEK293F *SIRT7*^{-/-} cells with SIRT7(K272A, K275A, K276A) to test its substrate selectivity within the endogenous environment of the cell nucleus. Here, we observed that, as indicated by our *in vitro* experiments, this mutant is much more active than wild type SIRT7 on H3K18ac targets, while its activity on H3K36ac remains similar. We have added Fig. 5E showing this effect:

Fig. 5. Loops flanking SIRT7 catalytic domain control substrate selectivity. (...) e Violin plot (truncated) of H3K18ac and H3K36ac levels in HEK293F *SIRT7*^{-/-} cells upon transient transfection with SIRT7-mCherry constructs as indicated, measured by immunofluorescence. Internal lines represent median and quartile values of $n \geq 3$ distinct experiments (ns: $p > 0.05$) and light background line indicates median value for WT SIRT7. See Supplementary Fig 13 for SIRT7 Western blots, representative microscopy images and control H4K16ac quantification.

we updated the results section with the following:

“Strikingly, mutating lysines K272, K275 and K276 in loop 2 to alanine resulted in even higher SIRT7 deacetylation activity on H3K18ac, leading to the inversion of substrate preference towards H3K18 over H3K36 (Fig. 5D). Indeed, when transfecting HEK293F *SIRT7*^{-/-} cells with this mutant, the nuclear H3K18ac levels decreased to a much larger extent than with wild type SIRT7, while H3K36ac remained at a similar level (Fig. 5E and Supplementary Fig. 13), which confirmed that SIRT7(K272A, K275A, K276A) is more active against the former modification and the overall inversion of SIRT7

substrate selectivity within the cellular environment. This mutant thus reveals that the loop between K272 and K276 is key for the substrate specificity of SIRT7.”

we added Supplementary Fig. 13 showing SIRT7 expression, cell images, and quantification of H4K16ac as control histone mark:

Supplementary Fig. 13. Substrate selectivity of SIRT7 mutants in cells. **A**, Western blot of HEK293F *SIRT7*^{-/-} lysates upon transfection and incubation with SIRT7-mCherry constructs as indicated. Since the commercial SIRT7 antibody binds an N-terminal peptide that is missing in SIRT7(40–400), an mCherry antibody was used to detect this construct instead. **B**, Spinning-disk confocal images of HEK293F *SIRT7*^{-/-} cells upon transient transfection and H3K18ac immunofluorescence. Scale bars represent 10 μ m. **C**, Images for H3K36ac immunofluorescence. **D**, Images for H4K16ac immunofluorescence, used here as control. **E**, Violin plot (truncated) of H4K16ac levels in transfected cells. Lines represent median and quartile values of $n \geq 3$ distinct experiments (ns: $p > 0.05$) and light background line indicates median value for WT SIRT7.

and we updated the Methods section in the main manuscript to include all new experiments.

All concerns provide above are based on trust of high rigor of all experiments. The authors may check some of the results to see if some of these confusing observations were due to experimental errors. Nevertheless, this is an important piece of study.